# CauchyNet: Compact and Data-Efficient Learning Using Holomorphic Activation Functions

**Hong-Kun Zhang** [* 1] **Xin Li** [* 2 3] **Sikun Yang** [4 3] **Zhihong Xia** [1]

## Abstract

CauchyNet is a compact complex-valued network whose hidden units form products of shifted reciprocal features, motivated by Cauchy-type kernel representations. The design targets regression problems with sharp rational-like spikes and partially observed inputs, where standard real-valued networks often require large width. We prove that finite linear combinations of multivariate Cauchy kernels are dense in $C(M)$ on compact $M \subset \mathbb{R}^N$ with $N$ being the input dimension, and that CauchyNet can realize these kernel sums via its complex biases and output weights. In experiments, CauchyNet reaches lower error on near-singular and gap-filling benchmarks, often with substantially fewer trainable parameters in our settings; results on smooth and piecewise-affine targets are mixed and delineate the method's intended regime.

## 1. Introduction

Modern sequence models routinely trade accuracy for scale, which is often incompatible with edge deployment, limited training data, or partially observed signals. In these regimes, standard real-valued architectures can become width-hungry when the target contains sharp spikes or rational-like structure, and they can degrade under missingness.

Targets with localized peaks or near-singular behavior expose two structural limits of standard architectures. A ReLU network needs at least $\Omega(1/\delta)$ piecewise-linear

pieces to resolve a peak whose support has width $\sim \delta$ around a complex pole at distance $\delta$, since each unit contributes only one linear bend. Sinusoidal networks (SIREN) capture smooth oscillations but produce Gibbs-style ringing near sharp transitions (Sitzmann et al., 2020; Vonderfecht & Liu, 2024). Transformer- and N-BEATS-style models are flexible but parameter-heavy, putting them out of reach of edge or data-scarce deployments.

**Construction.** Discretizing the multivariate Cauchy integral formula on the boundary of a complex neighborhood of the input domain yields kernel atoms $K(\boldsymbol{\xi}, \mathbf{x}) = \prod_{i=1}^{N}(\xi_i - x_i)^{-1}$. Each atom is exactly the computation performed by one hidden unit of a single-layer complex-valued network with no input weights—only shifted reciprocals. This identification connects function approximation directly to classical complex analysis and motivates a compact parameterization.

We introduce **CauchyNet**, a complex-valued single-hidden-layer architecture whose $k$-th hidden unit evaluates a product of shifted reciprocal terms. Each unit corresponds to a *pole location* in the complex plane, and the network output is a linear combination of these kernel-like atoms; when needed, an imaginary-part penalty can encourage nearly real-valued predictions.

To make the inductive bias concrete, consider the one-dimensional target

$$g(x) = \sin(3x) + \frac{4}{(x-0.5)^2 + 0.01}, \quad x \in [-1, 1],$$

which combines a smooth oscillation with a localized rational spike. Figure 1 compares CauchyNet against a width-matched ReLU FNN (labelled "FFN" in the figure legend) under identical training conditions; CauchyNet captures the spike with substantially smaller error.

**Contributions.** (i) We propose CauchyNet, an inversion-based complex network that directly parameterizes pole-like features for compact approximation. (ii) We prove a universal approximation result by showing density of finite Cauchy-kernel expansions in $C(M)$ on compact do-

---

[1]School of Sciences, Great Bay University, Dongguan 523000, China [2]College of Computer Science and Technology, Dongguan University of Technology, China [3]Great Bay Institute for Advanced Study, Great Bay University, Dongguan 523000, China [4]School of Computing and Information Technology, Great Bay University, Dongguan 523000, China. Correspondence to: Zhihong Xia <xiazh@gbu.edu.cn>.

*Proceedings of the $43^{rd}$ International Conference on Machine Learning*, Seoul, South Korea. PMLR 306, 2026. Copyright 2026 by the author(s).

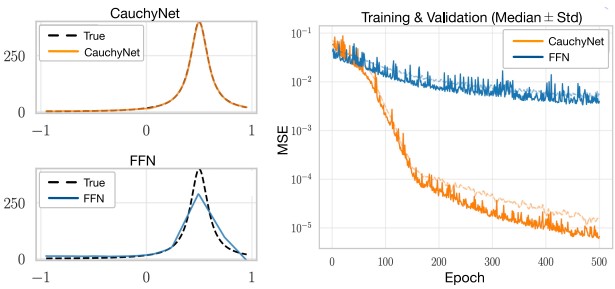

*Figure 1.* (Left) True function (dashed black line), CauchyNet (orange), and a width-matched ReLU feed-forward network (blue; labelled "FFN" in the legend, denoted FNN elsewhere in the paper and code) on a rational-spike target. CauchyNet tracks the steep peak near $x = 0.5$, while the FNN underestimates it. (Right) Training and validation loss trajectories (log scale) over 500 epochs. Shaded regions indicate standard deviation across 10 independent runs.

mains and connecting these expansions to the CauchyNet parameterization. (iii) We demonstrate gains on approximation, missing-value imputation, and time-series forecasting benchmarks under data scarcity.

## 2. Preliminaries

Cauchy's integral formula supplies the reciprocal factor that motivates the CauchyNet hidden units and the kernel approximation argument below. If $f$ is holomorphic on and within a closed contour $C \subset \mathbb{C}$, then for any $z$ inside $C$,

$$f(z) = \frac{1}{2\pi i} \oint_C \frac{f(\xi)}{\xi - z} d\xi.$$

This formula is driven by the inversion term $(\xi - z)^{-1}$, providing a basis for *kernel-based expansions*. Extending to higher dimensions (i.e., $\mathbb{C}^N$) involves products of such inversions. More precisely, let $U$ be an open set in the complex space, such that

$$U = \prod_{i=1}^N U_i \subset \mathbb{C}^N, \quad \text{with} \quad \bar{U} := \prod_{i=1}^N \bar{U}_i \subset \mathbb{C}^N,$$

where each $U_i$ is an open domain of $\mathbb{C}$, and $\bar{U}_i$ denotes its closure. The domain $U$ represents the Cartesian product of these $N$ sets, forming a multidimensional complex space. Let $M \subset U$. Suppose $f : M \to \mathbb{R}$ is extended to an analytic function $\bar{f}$ on $U$ and continuous on its closure $\bar{U}$. Then the high-dimensional Cauchy's integral formula is articulated for all $\boldsymbol{z} = (z_1, \cdots, z_N) \in \mathring{U}$:

$$\bar{f}(\boldsymbol{z}) = \frac{1}{(2\pi i)^N} \times \tag{1}$$

$$\int_{\zeta_1 \in \partial U_1} \cdots \int_{\zeta_N \in \partial U_N} \frac{\bar{f}(\zeta)}{(\zeta_1 - z_1) \cdots (\zeta_N - z_N)} d\zeta_1 \ldots d\zeta_N,$$

where $\zeta = (\zeta_1, \cdots, \zeta_N)$.

We use this reciprocal structure to define a Cauchy kernel on compact subsets of $\mathbb{R}^N$. Sampling boundary points $\{\boldsymbol{\xi}_k\}$ gives finite combinations of terms of the form $(\xi_k - x)^{-1}$; Sec. 5.3 proves that such finite sums are dense in the relevant continuous-function spaces. The neural activation uses the same inversion directly rather than building a piecewise-linear approximation to a rational feature. Rational functions have also been used effectively for complex-function approximation (Broomhead & Lowe, 1988).

## 3. Related Work

We focus on two issues from Section 1: parameter-efficient models for data-scarce regimes, and holomorphic tools for near-singular approximation and missing-data imputation. A detailed catalog of architectures appears in the supplement.

**Compact and efficient architectures.** Pruning and compression (Han et al., 2015; Iandola et al., 2016) can shrink mainstream models, yet typically inherit the inductive biases that struggle on sharp, rational near-singularities or severely incomplete data. Within forecasting and edge settings, methods such as N-BEATS and seasonal-trend decompositions (Sbrana & Lima de Castro, 2023; Bandara et al., 2025; Oreshkin et al., 2020) push parameter and compute efficiency, but rely on multi-block architectures.

**Approximation theory and rational kernels.** Rational expansions provide strong approximations near singularities (Broomhead & Lowe, 1988; Park & Sandberg, 1991). A real-valued rational network construction was proposed by Boullé, Nakatsukasa and Townsend (NeurIPS 2020): rational activations $P(x)/Q(x)$ are applied element-wise. Our design departs by applying $\prod_{i=1}^N z_i^{-1}$ *across all input dimensions jointly* in a single hidden unit, and by deriving the construction directly from the multivariate Cauchy integral formula rather than from learnable Padé coefficients.

**Sinusoidal and complex-valued networks.** Wavelets, radials, and sinusoidal activations (Sitzmann et al., 2020; Vonderfecht & Liu, 2024) encode localized features but are less suited to rational-like spikes; SIREN produces Gibbs-style oscillations near sharp transitions. Complex-valued networks (Hirose, 1992; Hammad, 2024) mostly rely on multi-layer architectures with high parameter counts. CauchyNet instead uses a single hidden layer with a reciprocal activation derived from *Cauchy's integral formula*, giving a compact holomorphic parameterization for near-singular targets.

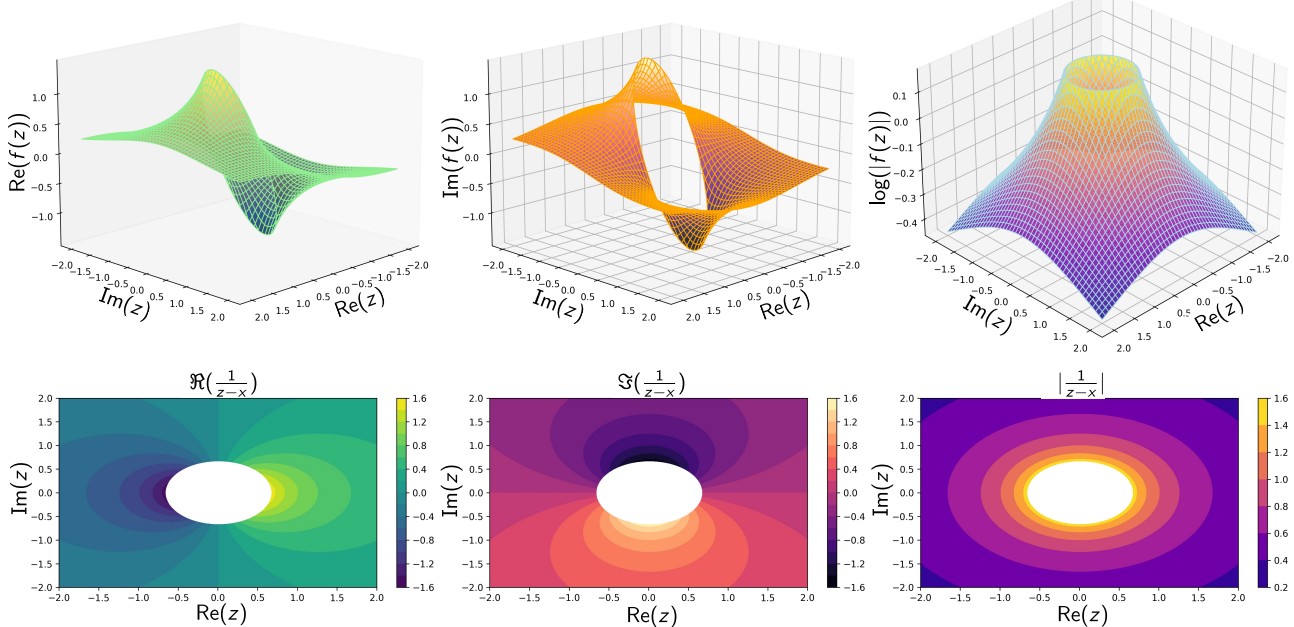

*Figure 2.* Cauchy activation function $\mathscr{X}(\boldsymbol{z})$ for $z$ values outside a small circular disc. (Top row) Real part, imaginary part, and magnitude of $\mathscr{X}(\boldsymbol{z})$. (Bottom row) Corresponding 2D contour plots. The white region marks the excluded circular disc in the complex plane.

## 4. The Proposed CauchyNet

CauchyNet is a single-hidden-layer complex-valued architecture built from a Cauchy-inspired reciprocal activation. The parameterization is intentionally small, with edge and data-scarce settings in mind, and is designed for oscillatory or near-singular targets.

### 4.1. Cauchy Activation Function

Inspired by *Cauchy's integral formula*, we use an inversion-based activation for near-singular and oscillatory targets (Li et al., 2025).

**Definition 1 (Cauchy activation function)** *Let* $\mathbb{C}_*^N := \{\boldsymbol{z} = (z_1, \ldots, z_N) \in \mathbb{C}^N : z_i \neq 0 \ \forall i = 1, \ldots, N\}$. *The Cauchy activation function* $\mathscr{X} : \mathbb{C}_*^N \to \mathbb{C}$ *is defined by*

$$\mathscr{X}(\boldsymbol{z}) = \prod_{i=1}^{N} z_i^{-1}.$$

The activation multiplies reciprocal coordinates and returns a single complex value. Its derivative is simple: when $\boldsymbol{z} \in \mathbb{C}$ is a single complex number,

$$\frac{d}{dz}\mathscr{X}(z) = -\frac{1}{z^2} = -\mathscr{X}(z)^2.$$

Fig. 2 illustrates the 3D plots of real, imaginary, and magnitude parts of the activation function $\mathscr{X}(\boldsymbol{z})$ for $z$ values taken outside of a small circular disc.

Because the reciprocal terms are multiplied across coordinates, a hidden unit can encode cross-dimensional interactions without an additional affine input map. Additive activations such as ReLU typically need depth or width to represent the same multiplicative structure.

The inversion-based activation is holomorphic away from its poles, which gives an analytic parameterization and supports Wirtinger-gradient training (Hirose, 1992). Its reciprocal form attenuates large shifted inputs and amplifies inputs near selected pole locations, matching the near-singular targets considered here.

### 4.2. Architecture and Parameterization

CauchyNet has four components: input embedding, complex bias shifts, the reciprocal activation, and a linear output combination. Unlike standard MLPs of the form $\sigma(Wx+b)$, CauchyNet uses *only* complex bias shifts, with no input weight matrix $W$. This is because the Cauchy kernel $1/(\xi - x)$ is translation-covariant: any input scaling $w$ is absorbable into a rescaled bias and an overall output coefficient, i.e., $1/(wx+b) = (1/w) \cdot 1/(x - (-b/w))$, so a preactivation weight would be strictly redundant. This observation underlies both the parameter count in Eq. 4 and the universality result in Theorem 2.

**Forward Pass and Output Layer.** The forward pass of CauchyNet proceeds through the following stages:
1) *Input Embedding into the Complex Plane*: An input vector $\boldsymbol{x} \in \mathbb{R}^N$ is embedded into the complex plane as $\boldsymbol{z} \in \mathbb{C}^N$

**Algorithm 1** Forward Pass of CauchyNet

---

**Require:** Input $\mathbf{x} \in \mathbb{R}^N$, Complex bias $\mathbf{B} \in \mathbb{C}^{h \times N}$, Complex coefficients $\mathbf{C} \in \mathbb{C}^h$, offset $\varepsilon > 0$
**Ensure:** Real output $y$, imaginary error $e$
 1: $\mathbf{z} \leftarrow \mathbf{x} + i\mathbf{0}$ {Embed into $\mathbb{C}^N$}
 2: $\mathbf{Z} \leftarrow$ Replicate $\mathbf{z}$ across $h$ rows
 3: $\mathbf{H} \leftarrow \mathbf{Z} + \mathbf{B}$ {Apply complex bias shifts}
 4: **for** $k = 1$ **to** $h$ **do**
 5: $\quad h_k \leftarrow \prod_{i=1}^{N}(H_{k,i} + \varepsilon)^{-1}$ {Cauchy Activation}
 6: $\mathbf{h} \leftarrow (h_1, \ldots, h_h)^{\top}$
 7: $o \leftarrow \mathbf{C}^{\top}\mathbf{h}/h$ {$1/h$ keeps output bounded as width grows}
 8: $y \leftarrow \Re(o)$
 9: $e \leftarrow \Im(o)$
10: **return** $(y, e)$

---

via

$$\mathbf{z} = \mathbf{x} + i\mathbf{0},$$

ensuring no imaginary components initially. This complex representation allows each hidden neuron to apply distinct bias shifts.

2) *Complex Bias Shifts*: The embedded input is replicated across $h$ hidden neurons, forming $\mathbf{Z} \in \mathbb{C}^{h \times N}$. A learnable complex bias matrix $\mathbf{B} \in \mathbb{C}^{h \times N}$ is then added:

$$\mathbf{H} = \mathbf{Z} + \mathbf{B},$$

where each row $\mathbf{H}_k \in \mathbb{C}^N$ corresponds to the $k$-th hidden neuron.

3) *Cauchy Activation*: Each hidden neuron applies the inversion-based activation function:

$$h_k = \prod_{i=1}^{N}(H_{k,i} + \varepsilon)^{-1}, \quad k = 1, \ldots, h.$$

This yields the activation vector $\mathbf{h} \in \mathbb{C}^h$.

4) *Output Combination*: The activated hidden units are aggregated using a complex coefficient vector $\mathbf{C} \in \mathbb{C}^{h \times 1}$, with a $1/h$ normalization to keep the output magnitude bounded independently of width:

$$o = \frac{1}{h} \mathbf{C}^{\top}\mathbf{h}, \tag{2}$$

resulting in a complex output $o = y + ie$, where $y = \Re(o)$ is the real-valued prediction, and $e = \Im(o)$ serves as an error term. The $1/h$ factor prevents the summation from growing with $h$ when individual kernel evaluations $h_k$ can be large (i.e., when poles sit close to the input domain); it is also what the released CauchyNet implementation uses.

**Backward Pass and Gradient Computation.** Algorithm 1 gives the forward pass. The backward pass uses

Wirtinger derivatives to compute gradients with respect to the complex parameters $\mathbf{B}$ and $\mathbf{C}$ by treating the real and imaginary parts as coupled real variables. Modern frameworks such as PyTorch handle the required complex-valued autograd operations directly.

---

**Algorithm 2** Backward Pass of CauchyNet

---

**Require:** Gradients of loss with respect to $y$ and $e$, Complex bias $\mathbf{B}$, Complex coefficients $\mathbf{C}$, Hidden activations $\mathbf{h} \in \mathbb{C}^h$, width $h$
**Ensure:** Gradients with respect to $\mathbf{B}$ and $\mathbf{C}$
 1: Compute $\frac{\partial \mathcal{L}}{\partial o^*} = \frac{1}{2}\left(\frac{\partial \mathcal{L}}{\partial y} + i\frac{\partial \mathcal{L}}{\partial e}\right)$
 2: Compute gradients w.r.t. $\mathbf{C}$: $\frac{\partial \mathcal{L}}{\partial \mathbf{C}_k} = \frac{1}{h} h_k \cdot \frac{\partial \mathcal{L}}{\partial o^*}, \quad k = 1, \ldots, h$
 3: **for** $k = 1$ **to** $h$ **do**
 4: $\quad$ Compute $\frac{\partial \mathcal{L}}{\partial h_k} = \frac{1}{h}\mathbf{C}_k \cdot \frac{\partial \mathcal{L}}{\partial o^*}$
 5: $\quad$ Compute $\frac{\partial \mathcal{L}}{\partial B_{k,i}} = -\frac{\partial \mathcal{L}}{\partial h_k} \cdot \prod_{\substack{j=1 \\ j \neq i}}^{N}(H_{k,j} + \varepsilon)^{-1} \cdot (H_{k,i} + \varepsilon)^{-2}, \quad \forall i = 1, \ldots, N$
 6: **return** Gradients w.r.t. $\mathbf{B}$ and $\mathbf{C}$

---

**Training Objective and Complexity.** Let $y_{\text{true}}$ denote the target output. The model produces a complex output $o = y + ie$, where $y$ is the real-valued prediction, and $e = \Im(o)$ is the imaginary error term, see Eq. 2. To enforce accurate real-valued predictions, we define the loss function as:

$$\mathcal{L} = (y - y_{\text{true}})^2 + \lambda|e|^2, \tag{3}$$

where $\lambda \geq 0$ is a task-level hyperparameter. Positive values penalize the magnitude of the imaginary component and encourage $e \approx 0$ during training; $\lambda = 0$ recovers the pure real-output loss used by the final fixed-pole gap-filling configuration.

Using Wirtinger derivatives, the loss gradients with respect to $\mathbf{B}$ and $\mathbf{C}$ can be passed directly to standard gradient-based optimizers. Modern frameworks such as PyTorch and TensorFlow support these complex-valued backpropagation operations.

Each forward and backward pass of CauchyNet incurs computational complexity $\mathcal{O}(hN)$, where $h$ is the number of hidden units and $N$ is the input dimensionality, as detailed in Eq. 4.

$$\underbrace{2(h \times N)}_{\text{Complex Biases } B} + \underbrace{2h}_{\text{Complex Coefficients} \Theta} = \underbrace{2h(N+1)}_{\text{(Real Parameters)}}. \tag{4}$$

This cost and parameter count are smaller than those of recurrent or attention-based architectures in the one-dimensional settings considered here, especially when $h$ and $N$ are moderate or inference must run on limited hardware.

**Real-Output Variant via Conjugate-Symmetric Poles.**
When the planar domains $U_i$ are chosen symmetric about the real axis—as we assume throughout Sec. 5.3—the imaginary-part penalty can be eliminated entirely by enforcing conjugate-symmetric hidden units. Concretely, organize the $h$ hidden units into $h/2$ pairs and constrain each pair $(k, k')$ by

$$\mathbf{B}_{k',i} = \overline{\mathbf{B}_{k,i}}, \quad \mathbf{C}_{k'} = \overline{\mathbf{C}_k}, \quad i = 1, \ldots, N.$$

For any real input $\mathbf{x} \in \mathbb{R}^N$, the paired contribution is

$$\mathbf{C}_k h_k + \mathbf{C}_{k'} h_{k'} = 2 \Re(\mathbf{C}_k h_k) \in \mathbb{R},$$

so $\Im(o) \equiv 0$ by construction and the training loss collapses to pure mean-squared error,

$$\mathcal{L}_{\text{pair}} = (y - y_{\text{true}})^2.$$

This variant removes the hyperparameter $\lambda$, halves the number of independent pole parameters, and still inherits the universality result of Theorem 2: any kernel sum $\sum_k \theta_k K(\boldsymbol{\xi}_k, \cdot)$ realizing Theorem 1 on a contour $\Gamma$ symmetric about $\mathbb{R}^N$ can be symmetrized into conjugate pairs $(\xi_k, \bar{\xi}_k)$ with coefficients $(\theta_k, \bar{\theta}_k)$ without loss of approximation power. In the experiments we use the unconstrained objective with $\lambda$ selected by the corresponding setup: the trainable-pole ablations use positive penalties, while the final fixed-pole gap-filling sweep selects $\lambda_{\text{imag}} = 0$. The conjugate-symmetric variant is a strictly simpler special case with the same expressive power.

**Parameter Initialization Strategies.** We provide two strategies for parameter initialization. The first one is rather general. The complex biases $\mathbf{B} \in \mathbb{C}^{h \times N}$ and complex coefficients $\mathbf{C} \in \mathbb{C}^{h \times 1}$ are initialized using a variant of the Xavier (Glorot) initialization scheme adapted for complex-valued parameters. Specifically, both the real and imaginary components are sampled from a normal distribution with zero mean and variance $\frac{2}{N+h}$:

$$\mathbf{B}_{k,i} \sim \mathcal{N}\left(0, \frac{2}{N+h}\right) + i\mathcal{N}\left(0, \frac{2}{N+h}\right), \tag{5}$$

$$\mathbf{C}_k \sim \mathcal{N}\left(0, \frac{2}{N+h}\right) + i\mathcal{N}\left(0, \frac{2}{N+h}\right), \quad \forall k, i \tag{6}$$

This Xavier variant controls the initial scale of the complex activations. Inspired by *Cauchy's integral formula* (Eq. 1), we can alternatively use an *elliptical initialization* that places each pole on an ellipse in the complex plane symmetric about the real axis:

$$\mathbf{B}_{k,i} = r_{\text{re}} \cos\theta_{k,i} + i\, r_{\text{im}} \sin\theta_{k,i}, \quad \theta_{k,i} \sim \text{Uniform}[0, 2\pi].$$

The radii $(r_{\text{re}}, r_{\text{im}})$ should match the input scale: for inputs normalized to $[-1, 1]$ we use $(r_{\text{re}}, r_{\text{im}}) = (1.5, 0.5)$

throughout our experiments, which places poles right around but outside the input domain so that $1/(\mathbf{B}_{k,i} - x_i)$ varies meaningfully across $x_i \in [-1, 1]$ while remaining bounded. See Experiment 5 in the Supplement for an ablation of these radii.

# 5. Theoretical Analysis

We next prove that CauchyNet can approximate continuous functions on compact subsets of $\mathbb{R}^N$. The argument uses a Cauchy-type kernel built from the reciprocal factors in *Cauchy's integral formula*. Fig. 3 summarizes the derivation.

**Formal vocabulary.** We first pin down what we mean by "sharp spikes" and "near-singularities" via the analytic structure of the target.

**Definition 2 ($\delta$-near-singularity)** *Let $M \subset \mathbb{R}^N$ be compact and $f \in C(M; \mathbb{R})$. We say $f$ is $\delta$-near-singular on $M$ if $f$ admits a meromorphic extension $\bar{f}$ to a complex neighborhood $U \supset M$ with at least one pole $p$ satisfying $\text{dist}(p, M) < \delta$. Smaller $\delta$ corresponds to sharper peaks on $M$.*

**Definition 3 (Sharp spike)** *A* sharp spike *of $f$ is a localized region $V \subset M$ where $\|\nabla f\| \gg 1$. If $f$ has a simple pole at distance $\delta$ from $M$, then $\|\nabla f\| = \Omega(\delta^{-2})$ on the projection of $V$ near that pole; higher-order or coalescing poles strengthen the lower bound (e.g. to $\Omega(\delta^{-3})$ for a double pole).*

For example, $4/((x - 0.5)^2 + 0.01)$ factors as $4/[(x - 0.5 - 0.1i)(x - 0.5 + 0.1i)]$, hence has two simple poles at $x = 0.5 \pm 0.1\,i$ and is 0.1-near-singular on $[-1, 1]$ in the sense of Definition 2. These definitions make CauchyNet's structural advantage precise: its complex biases directly parameterize pole positions, so a single hidden unit per pole suffices to represent a $\delta$-near-singular target, whereas a ReLU network requires $\Omega(\delta^{-1})$ piecewise-linear pieces to cover the peak's $\sim \delta$-wide support with comparable error (since each ReLU unit contributes a single linear bend).

## 5.1. Problem Setup and Function Spaces

Let

$$M = \prod_{i=1}^{N} M_i \subset \mathbb{R}^N$$

be a compact domain (with each $M_i \subset \mathbb{R}$), and define

$$\mathcal{F}_{M,D} := C^0(M, D) = \{f : M \to D \mid f \text{ is continuous}\},$$

where $D = \mathbb{R}$ or $D = \mathbb{C}$. Given a set of observed training data

$$\mathcal{T} := \{(\mathbf{x}_i, \mathbf{y}_i)\}_{i=1}^{n},$$

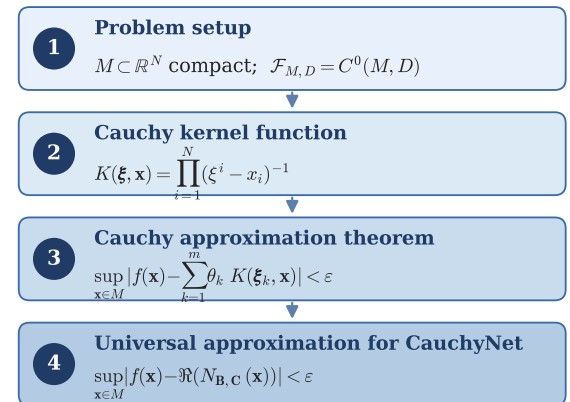

**Problem setup**

$M \subset \mathbb{R}^N$ compact; $\mathcal{F}_{M,D} = C^0(M, D)$

**Cauchy kernel function**

$K(\boldsymbol{\xi}, \mathbf{x}) = \prod_{i=1}^{N} (\xi^i - x_i)^{-1}$

**Cauchy approximation theorem**

$\sup_{\mathbf{x} \in M} |f(\mathbf{x}) - \sum_{k=1}^{m} \theta_k\, K(\boldsymbol{\xi}_k, \mathbf{x})| < \varepsilon$

**Universal approximation for CauchyNet**

$\sup_{\mathbf{x} \in M} |f(\mathbf{x}) - \Re(N_{\mathbf{B},\mathbf{C}}(\mathbf{x}))| < \varepsilon$

*Figure 3.* Roadmap of the theoretical analysis: four stages that take a continuous target $f$ on a compact domain $M$ to a CauchyNet realization $N_{\mathbf{B},\mathbf{C}}$ approximating $f$ to arbitrary accuracy.

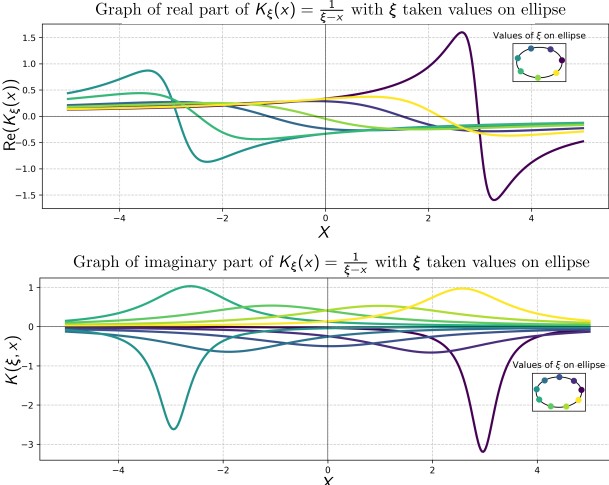

*Figure 4.* One-dimensional complex Cauchy kernel $K(\xi, x) = \frac{1}{\xi - x}$ for $\boldsymbol{\xi}$ values on an ellipse in the complex plane. The main plots show the real (top) and imaginary (bottom) parts of $K(\boldsymbol{\xi}, x)$ for $x \in [-5, 5]$. The inset shows the pole locations on an ellipse with semi-major and semi-minor axes of 6 and 2.

with $\mathbf{y}_i = f(\mathbf{x}_i)$ for an unknown dynamical system $f \in \mathcal{F}_{M,\mathbb{R}}$, our goal is to find a complex-valued function $g \in \mathcal{F}_{M,\mathbb{C}}$ that approximates $f$ on these samples. In practice, we minimize an empirical loss such as

$$L_{f,T}(g) := \frac{1}{n} \sum_{(\mathbf{x}_i, \mathbf{y}_i) \in \mathcal{T}} \left( |\Re(g(\mathbf{x}_i)) - \mathbf{y}_i|^2 + \lambda \left| \Im(g(\mathbf{x}_i)) \right|^2 \right),$$

where $\Re(g(\mathbf{x}))$ denotes the real part of $g(\mathbf{x})$, and $\Im(g(\mathbf{x}))$ denotes the imaginary part. For $\lambda > 0$ this loss encourages $g$ to produce nearly real-valued predictions; for $\lambda = 0$ it trains only the real part of the complex predictor.

### 5.2. Cauchy Kernel Function

To bridge real-valued approximation with holomorphic kernels, we embed $M$ into an open domain

$$U = \prod_{i=1}^{N} U_i \subset \mathbb{C}^N,$$

with each $U_i$ chosen symmetric about the real axis and containing $M_i$. We will use the product contour $\Gamma = \partial U_1 \times \cdots \times \partial U_N$, which is the natural boundary set appearing in the multivariate Cauchy integral formula on product domains.

**Definition 4 (Cauchy kernel function)** *Let $M = \prod_{i=1}^{N} M_i \subset \mathbb{R}^N$ be compact and choose bounded planar domains $U_i \subset \mathbb{C}$ such that $M_i \subset U_i$ and $\partial U_i$ is a positively oriented piecewise-$C^1$ Jordan curve. Define the product domain $U := \prod_{i=1}^{N} U_i \subset \mathbb{C}^N$ and the product contour $\Gamma := \partial U_1 \times \cdots \times \partial U_N$. For $\boldsymbol{\xi} = (\xi^1, \ldots, \xi^N) \in \Gamma$ and $\mathbf{x} = (x_1, \ldots, x_N) \in M$, define $K(\boldsymbol{\xi}, \mathbf{x}) := \prod_{i=1}^{N} \frac{1}{\xi^i - x_i}$.*

Since $M$ is compact, and lies strictly within $U$, every denominator $\xi_k^i - x_i$ is bounded away from zero, so $K(\boldsymbol{\xi}_k, \mathbf{x})$ is uniformly bounded. Small changes in $\mathbf{x}$ then produce smooth changes in $K(\boldsymbol{\xi}_k, \mathbf{x})$. The kernel has the same reciprocal form as the CauchyNet activation in Sec. 4.

Fig. 4 illustrates the behavior of the real and imaginary parts of $K(\boldsymbol{\xi}, \cdot)$, for different $\boldsymbol{\xi}$ values along an ellipse in the complex plane.

### 5.3. Universal Approximation for Cauchy Kernels

Let $M = \prod_{i=1}^{N} M_i \subset \mathbb{R}^N$ be compact. Choose bounded planar domains $U_i \subset \mathbb{C}$ such that $M_i \subset U_i$ and $\partial U_i$ is a positively oriented piecewise-$C^1$ Jordan curve. Define the product domain $U := \prod_{i=1}^{N} U_i \subset \mathbb{C}^N$ and the product contour

$$\Gamma := \partial U_1 \times \cdots \times \partial U_N.$$

For $\boldsymbol{\xi} = (\xi^1, \ldots, \xi^N) \in \Gamma$ and $\mathbf{x} = (x_1, \ldots, x_N) \in M$, define the Cauchy kernel

$$K(\boldsymbol{\xi}, \mathbf{x}) := \prod_{i=1}^{N} \frac{1}{\xi^i - x_i}.$$

Let $\delta_i := \inf\{|\zeta - x| : \zeta \in \partial U_i,\ x \in M_i\} > 0$ and $\delta := \min_i \delta_i$. The following theorem states the resulting density property.

**Theorem 1 (Cauchy-kernel universality on $C(M)$)** *For any $f \in C(M; \mathbb{R})$ and any $\varepsilon > 0$, there exist $m \in \mathbb{N}$, nodes $\boldsymbol{\xi}_1, \ldots, \boldsymbol{\xi}_m \in \Gamma$ and weights $\theta_1, \ldots, \theta_m \in \mathbb{C}$ such*

*that*

$$\sup_{\mathbf{x} \in M} \left| f(\mathbf{x}) - \sum_{k=1}^{m} \theta_k \, K(\boldsymbol{\xi}_k, \mathbf{x}) \right| < \varepsilon.$$

*Equivalently, finite linear combinations of $\{K(\boldsymbol{\xi}, \cdot) : \boldsymbol{\xi} \in \Gamma\}$ are dense in $C(M; \mathbb{R})$.*

Full details of the proof are given in the supplement.

The proof is constructive (via tensor-product quadrature on $\Gamma$), which yields a quantitative rate that scales with the analytic regularity of the target.

**Corollary 1 (Approximation rates)** *Let $h$ denote the number of hidden units of a CauchyNet on a product contour $\Gamma$ with $m = h^{1/N}$ nodes per coordinate.*

1. ***Analytic targets.*** *If $f$ admits a holomorphic extension to a neighborhood of $\bar{U}$ with analyticity strip width $d > 0$, then $\sup_{\mathbf{x} \in M} \left| f(\mathbf{x}) - \Re(N_{\mathbf{B},\mathbf{C}}(\mathbf{x})) \right| = \mathcal{O}\left( \exp(-2\pi d \, h^{1/N}) \right).$*

2. *$C^k$ **targets.** If $f \in C^k(M; \mathbb{R})$, then $\sup_{\mathbf{x} \in M} \left| f(\mathbf{x}) - \Re(N_{\mathbf{B},\mathbf{C}}(\mathbf{x})) \right| = \mathcal{O}\left( h^{-k/N} \right).$*

The analytic case follows from the exponential convergence of the trapezoidal rule for periodic analytic integrands on $\Gamma$ (Trefethen & Weideman, 2014); the $C^k$ case combines a Jackson-type polynomial approximation step with the analytic rate applied to the polynomial. Full details and constants are in the supplement.

Choosing enough contour points on $\Gamma$ and appropriate weights $\theta_k$ therefore gives a uniform approximation to any continuous function on $M$. Because CauchyNet learns complex biases analogous to the $\boldsymbol{\xi}_k$ and an output vector analogous to $\theta_k$, it inherits the following universal approximation property. Let $C_{\text{net}}$ denote the collection of all CauchyNets.

**Theorem 2 (Universal approximation for CauchyNet)**
*Let $f \in C(M; \mathbb{R})$ and $\varepsilon > 0$. There exist $h \in \mathbb{N}$ and parameters $(\mathbf{B}, \mathbf{C})$ for a width-$h$ CauchyNet $N_{\mathbf{B},\mathbf{C}}$ such that*

$$\sup_{\mathbf{x} \in M} \left| f(\mathbf{x}) - \Re(N_{\mathbf{B},\mathbf{C}}(\mathbf{x})) \right| < \varepsilon.$$

Proof. By Theorem 1, $f$ can be approximated by a finite kernel sum $\sum_{k=1}^{h} \theta_k K(\boldsymbol{\xi}_k, \mathbf{x})$. Choose $\mathbf{B}$ so that each hidden unit implements the corresponding pole locations (up to the constant factor induced by the sign convention in Sec. 4), and set $\mathbf{C}$ to match the complex weights. See the supplement for the explicit parameter mapping.

*Remark (Bias-only universality).* Theorem 2 does not require an input weight matrix $W$. Universality is achieved using only complex biases $\mathbf{B}$ and output coefficients $\mathbf{C}$. In contrast, ReLU/Tanh universal approximation intrinsically requires an affine preactivation $Wx + b$: without $W$, a single ReLU hidden unit reduces to a shifted rectifier along a fixed axis and cannot form a dense family in $C(M)$. The Cauchy kernel avoids this because its translation covariance, $1/(wx + b) = (1/w) \cdot 1/(x - (-b/w))$, makes the scaling $w$ absorbable into a rescaled bias $\xi = -b/w$ together with an overall output factor, so bias-only is already universal. This is the precise origin of the $2h(N+1)$ real-parameter count asserted in Eq. 4, and it is one of the two properties (together with holomorphy) that distinguish CauchyNet from real-valued MLP baselines at matched width.

Theorem 2 shows that for any continuous function $f$ and any $\varepsilon > 0$, some CauchyNet in $C_{\text{net}}$ approximates $f$ to within $\varepsilon$. A width-$h$ CauchyNet uses about $2\,h\,(N+1)$ real parameters and has computational complexity $\mathcal{O}(h\,N)$. In the experiments, the best regularization choice depends on whether pole locations are fixed or trainable. We next test the model on approximation, forecasting, and missing-value imputation tasks (Sec. 6 and the supplement).

## 6. Experiments

The experiments focus on data-scarce settings, small parameter budgets, near-singular targets, and missing-value imputation. The supplement gives the full protocols and additional results.

### Experiment 1: Function Approximation with Sharp Peaks under Limited Data.

We first test a one-dimensional function with a rational spike, a smooth Gaussian term, and a discontinuous oscillatory component. The target function $f(x)$ on $[-1, 1]$ is:

$$f(x) = \frac{1}{(x + 0.6)^2 + 0.005} - 40\, e^{-2(x+0.4)^2}$$
$$+ \; 50 \operatorname{sign}(x) \, |\sin(3x) + 0.8|^{1.5} \, \sin(10x).$$

This function combines a sharp rational spike near $x = -0.6$ (from the first term, with poles at $-0.6 \pm \sqrt{0.005}\, i$), a smooth Gaussian dip near $x = -0.4$, and a discontinuous sign jump at $x = 0$ together with high-frequency oscillations from $\sin(10x)$. To simulate data scarcity in the qualitative visualization, 150 evenly spaced points are used for training, with 75 for validation and 75 for testing. We trained the compared networks for 200 epochs. A separate released trainable-pole, parameter-matched small-$n$ run on this mixed target is effectively tied with a ReLU FNN (test MSE 0.041 vs. 0.040), so Fig. 1 should be read as qualitative spike-fitting context for that legacy optimizer setting. The supplement adds a fixed-pole coefficient-fitting variant on the same target: with $h{=}64$ and only 128 learned real coefficients, ridge-fitted CauchyNet reaches dense-grid MSE

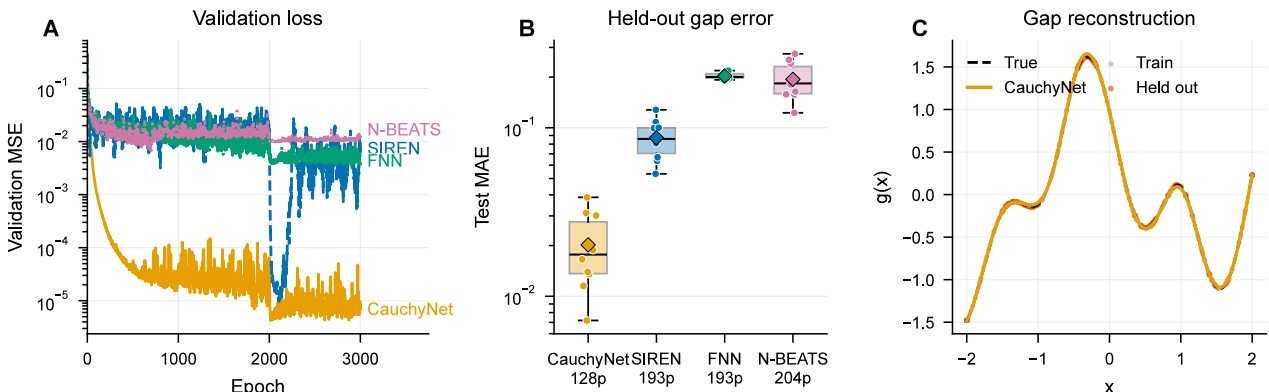

*Figure 5.* Final one-dimensional gap-filling results generated from the saved final-result files. (a) Mean validation loss over 10 seeds at the selected configuration. (b) Per-seed test MAE on held-out turning-point neighborhoods; diamonds mark means and model labels give trainable parameter counts. (c) Representative CauchyNet reconstruction from the same final configuration, with held-out samples concentrated in the gap regions.

$0.0027 \pm 0.0020$ at $n_{\text{train}}{=}20$, versus $0.0403 \pm 0.0194$ for a 256-parameter ReLU FNN.

CauchyNet captures localized rational structure with a compact parameterization, but the mixed-target result also clarifies an optimization boundary: trainable pole locations can be fragile at small sample sizes, while a fixed Cauchy dictionary with closed-form coefficient fitting is much more stable. The following experiments therefore separate pole-like cases, fixed-dictionary variants, and limitation cases.

**Experiment 2: Imputation of Missing Values Using CauchyNet**

We test CauchyNet's capacity to impute missing data in one-dimensional settings, especially in zones with steep gradients or near-singularities.

**One-Dimensional Gap-Filling.** We define the target function $g(x)$ on $[-2, 2]$ combining trigonometric, polynomial, and rational components:

$$g(x) = \sin(2x - 4) + 0.5\cos(5x - 5) + \frac{0.05}{(x - 1)^2 + 0.1}$$
$$+ \frac{0.01}{(x + 0.5)^2 + 0.05} - 0.01(x^2 - x^3).$$

This function has turning-point neighborhoods in which interpolation is deliberately difficult. In the final reproducible split, 200 training and 50 validation samples are drawn away from these neighborhoods, while 100 held-out test samples are drawn within distance $0.15$ of a turning point. All models in Fig. 5 use the same training budget of 3,000 epochs and the same learning rate.

CauchyNet accurately reconstructs the withheld regions, whereas baselines such as SIREN, N-BEATS, and FNN ex-

hibit larger held-out errors under the same budget. Fig. 5(a) shows the validation dynamics, Fig. 5(b) reports per-seed held-out MAE, Fig. 5(c) visualizes a representative reconstruction in the missing regions, and Table 1 gives the corresponding aggregate test errors.

**Experiment 3: Additional Empirical Evidence**

Several additional studies clarify the empirical scope of the method. All comparisons use parameter-matched single-layer baselines unless noted, with 10 seeds per cell; full protocols, tables, and additional ablations are in the supplement.

**Interpretation.** The advantage grows where the target's analytic structure aligns with the model's inductive bias: parameter-matched CauchyNet beats ReLU by $20\times$ on near-singular targets and by up to $102\times$ at moderate pole distances (Def. 2), while the fixed-pole ridge variant improves the original mixed 1D target by $14.8\times$ at $n_{\text{train}}{=}20$ with half the learned parameters. The wins are not universal: standard baselines are competitive on simple smooth targets, and ReLU wins by $1.6\times$ on the piecewise-affine "step-ramp" case where a piecewise-linear model is structurally well matched. The hybrid FNN$\rightarrow$Cauchy result extends the tests beyond 1D/2D: a small ReLU bottleneck handles dimensionality reduction, and CauchyNet then captures the pole-like structure in the reduced space. The Skip variant shows that multi-layer composition does not break theoretical guarantees (Cor. 1 bounds capacity, not training loss), and improves accuracy by $1.7\times$. The gap-filling study uses the CauchyNet configuration identified by a coordinate sweep over (lr, $h$, $n_{\text{train}}$, epochs) with all baselines re-trained at the same budget and 10 seeds: $h{=}64$, fixed elliptical poles $(r_{\text{re}}, r_{\text{im}}){=}(2.5, 0.4)$, $\lambda_{\text{imag}}{=}0$, lr$=5\times10^{-2}$, $n_{\text{train}}{=}200$, 3000 epochs. CauchyNet attains

*Table 1.* Final gap-filling configuration and test errors from the saved result files used to generate Fig. 5. Values are over 10 seeds; lower MAE is better.

| Setting | Value |
|---|---|
| Hidden size $h$ | 64 |
| Pole ellipse $(r_{re}, r_{im})$ | $(2.5, 0.4)$ |
| Trainable poles | no |
| Learning rate | $5 \times 10^{-2}$ |
| $\lambda_{imag}$ | 0 |
| Epochs / seeds | 3000/10 |

| Model | Mean MAE | Median | Max | Params |
|---|---|---|---|---|
| CauchyNet | **0.0202** | **0.0124** | **0.1618** | **128** |
| SIREN | 0.0872 | 0.0724 | 0.3349 | 193 |
| FNN | 0.2032 | 0.1962 | 0.5093 | 193 |
| N-BEATS | 0.1940 | 0.1860 | 0.8401 | 204 |

MAE 0.020, i.e. $4.3\times$ lower than SIREN, $9.6\times$ lower than a standard 3-block N-BEATS, and $10.1\times$ lower than a ReLU FNN, at 34% fewer trainable parameters (128 vs. 193–204).

**Empirical scope.** The released-result files give a consistent boundary for these claims. On the mixed target, the Adam-trained trainable-pole run is a tie (MSE 0.041 vs. 0.040 for ReLU), so the paper does not present it as a win; isolating the same Cauchy dictionary with fixed poles and ridge-fitted coefficients wins all small-$n$ cells, from $14.4\times$ to $81.7\times$. The non-smooth suite is also mixed but interpretable: CauchyNet wins on chirp, piecewise-smooth, and Gibbs targets, whereas ReLU wins by $1.6\times$ on the step-ramp target. On tabular data, standalone CauchyNet is not uniformly best; the released Hybrid FNN→Cauchy configuration has the strongest mean MSE on Diabetes, California Housing, and Wine.

## 7. Conclusion and Future Work

We introduced CauchyNet, a compact complex-valued network whose hidden units implement multivariate Cauchy kernel atoms derived from *Cauchy's integral formula*. The holomorphic parameterization works well on controlled near-singular targets, the fixed-pole mixed 1D study, and the final 1D gap-filling benchmark, while the supplement records mixed behavior on smooth, piecewise-affine, and standalone tabular settings.

**Limitations.** CauchyNet is not a uniformly dominant replacement for real-valued baselines. Its strongest and most consistent gains appear on near-singular, gap-filling, and fixed-pole dictionary settings; margins shrink or reverse when the target is better matched by a standard real activation. For example, the ReLU FNN wins the step-ramp case by $1.6\times$, and in the $\delta$-sweep the CauchyNet advantage collapses when the pole is extremely close to the real axis ($\delta = 0.02$ or 0.01), where the sampled data under-resolve the spike.

**Future work.** Open directions include higher input dimensions, longer sequences, deeper and residual variants, regularisation for complex parameters, and integration with neural operators or PDE solvers for problems with localised singularities. Extensions to generative settings (Song et al., 2021; Ho et al., 2020; Lou et al., 2024) remain unexplored.

## Impact Statement

The method studied here is a compact, data-efficient neural architecture grounded in complex analysis. Its smaller parameter counts and low computational cost may be useful in edge-computing or low-resource settings.

As with most general-purpose learning algorithms, the techniques presented here could be applied in a variety of downstream domains. However, this paper does not propose or evaluate any application involving sensitive personal data or automated decision-making systems. We do not identify any specific ethical risks unique to this work beyond those generally associated with the broader use of machine learning technologies.

## Acknowledgments

H-K. Z. is partially supported by Dongguan Key Laboratory for AI and Dynamical Systems, the National Natural Science Foundation of China (Grant No. 12571196), and Guangdong Basicand Applied Basic Research Foundation (Grant No. 2025A1515011952). Sikun Yang is supported by National Natural Science Foundation of China (NSFC) (Grant No. 62476047), Peking University Mathematics Challenge Funding Program (Grant No. 2024SRMC10), the Guangdong-Dongguan Joint Research Fund (Grant No. 2025A1515140098), and Dongguan Key Laboratory for AI and Dynamical Systems.

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

## A. Common Neural Architectures and Their Characteristics

Fig. 6 summarizes the neural architectures used as context for the experiments, with emphasis on parameter count, data requirements, and common failure modes in small-data settings.

## B. Notations and Symbols

Table 2 presents the key mathematical notations, symbols and the detailed descriptions.

*Table 2.* Key mathematical symbols (main paper notation).

| Symbol | Definition | Meaning |
|---|---|---|
| $M$ | $M = \prod_{i=1}^{N} M_i \subset \mathbb{R}^N$ | Compact real domain. |
| $U_i$ | $M_i \subset U_i \subset \mathbb{C}$ | Planar complex domain enclosing $M_i$. |
| $U$ | $U = \prod_{i=1}^{N} U_i \subset \mathbb{C}^N$ | Product domain. |
| $\Gamma$ | $\Gamma = \partial U_1 \times \cdots \times \partial U_N$ | Product contour for multivariate Cauchy formula. |
| $K(\boldsymbol{\xi}, \mathbf{x})$ | $\prod_{i=1}^{N} (\xi^i - x_i)^{-1}$ | Multivariate Cauchy kernel atom. |
| $f$ | $f \in C(M; \mathbb{R})$ | Target function to approximate. |
| $\boldsymbol{\xi}_k$ | $\boldsymbol{\xi}_k \in \Gamma$ | Pole locations for kernel atoms. |
| $\theta_k$ | $\theta_k \in \mathbb{C}$ | Complex weights in the kernel expansion. |
| $N_{\mathbf{B},\mathbf{C}}$ | CauchyNet | Network with parameters $\mathbf{B} \in \mathbb{C}^{h \times N}$, $\mathbf{C} \in \mathbb{C}^h$. |
| $h$ | integer | Hidden-layer width. |
| $\lambda$ | $\lambda \geq 0$ | Imaginary-part penalty weight. |

## C. Experiments

We report the experimental setup and additional results.

**Common Setup and Metrics.** Models are evaluated using Mean Squared Error (MSE), Mean Absolute Error (MAE), training time and number of parameters. Table 3 gives the default setup for the legacy one-dimensional trainable-pole runs; later final-result experiments state their deviations explicitly, including the final fixed-pole gap-filling configuration.

*Table 3.* Default training setup for legacy one-dimensional trainable-pole experiments.

| Parameter | Value |
|---|---|
| Data Split Ratio | 50% training, 25% validation, 25% testing |
| Architecture | One hidden layer with 128 neurons |
| Batch Size | 32 |
| Optimizer | Adam, learning rate 0.01, decayed by factor 0.5 every 100 epochs |
| Weight Decay | $1 \times 10^{-4}$ |
| Preprocessing | Target values normalized via `MinMaxScaler` |
| Random Seed | Fixed at 10 for reproducibility |
| Imaginary-Part Penalty | $\lambda = 0.1$ for default trainable-pole runs; final fixed-pole gap-filling uses $\lambda_{\text{imag}} = 0$ |

For the legacy one-dimensional parameter-count comparison, each baseline uses the architectural choice listed below at width $h = 128$. As shown in Table 4, CauchyNet is much smaller than recurrent and attention baselines, while SIREN, N-BEATS, and RBF have similar parameter counts at the same width.

*Table 4.* Number of parameters for each model on the 1D synthetic task (input = 1, output = 1). All counts are obtained by instantiating the released PyTorch architectures with width $h = 128$ and summing `p.numel()` over the parameters (complex tensors counted as two real scalars). See "Per-model configurations" below for the architectural details that determine each count.

| Model | CauchyNet | SIREN | N-BEATS | RBF | LSTM | Transformer | Informer |
|---|---|---|---|---|---|---|---|
| #Params | 512 | 385 | 385 | 385 | 67,201 | 132,993 | 199,553 |

**Counting convention.** CauchyNet uses complex parameters, counted as two real scalars. In the 1D synthetic setting with width $h = 128$, $(B, C) \in \mathbb{C}^{128 \times 1} \times \mathbb{C}^{128}$ gives $2h(N+1) = 512$ real parameters.

- LSTM (Hochreiter & Schmidhuber, 1997)

  **Strengths**: Effectively learns long-term temporal dependencies via gating.

  **Weaknesses**: Large parameter footprint and sensitive hyperparameters, especially in sparse-data contexts.

- Transformer (Vaswani et al., 2017)

  **Strengths**: Captures long-range correlations efficiently with abundant data.

  **Weaknesses**: Quadratic complexity in sequence length; memory-intensive.

- Informer (Zhou et al., 2021)

  **Strengths**: Employs sparse self-attention to enhance scalability for long sequences.

  **Weaknesses**: Multi-layer stacks involve thousands of parameters and require large datasets.

- N-BEATS (Oreshkin et al., 2020)

  **Strengths**: Provides accurate forecasting with interpretable decompositions into trend and seasonal components.

  **Weaknesses**: High parameter count and data-hungry; may overfit in scenarios with limited data.

- MLP

  **Strengths**: Simple and straightforward to implement and train.

  **Weaknesses**: Lacks specialized inductive biases for oscillatory or near-singular data.

- RBF Network (Park & Sandberg, 1991)

  **Strengths**: Excels at local approximations and can model a broad class of functions.

  **Weaknesses**: Placement of kernel centers is non-trivial; struggles with sharp rational spikes or steep gradients.

- SIREN (Sitzmann et al., 2020)

  **Strengths**: Sinusoidal activations capture smooth and high-frequency signals effectively.

  **Weaknesses**: Requires large capacity to model abrupt spikes or extremely limited data; not explicitly designed for missing-data imputation.

- TCN (Bai et al., 2018)

  **Strengths**: Uses dilated convolutions for efficient sequence modeling.

  **Weaknesses**: Deeper architectures increase parameter counts and can be computationally slow.

*Figure 6.* Common Neural Architectures and Their Characteristics

**Per-model configurations (1D, $h = 128$).** For reproducibility, the architectural choices that yield the counts in Table 4 are:

- **CauchyNet**: $\mathbf{B} \in \mathbb{C}^{h \times 1}$, $\mathbf{C} \in \mathbb{C}^{h}$; $2h(N+1) = 512$ real params.

- **SIREN**: Linear$(1 \rightarrow h)$ + sine + Linear$(h \rightarrow 1)$; $(h+h) + (h+1) = 385$.

- **N-BEATS**: single block of Linear$(1 \rightarrow h)$ + ReLU + Linear$(h \rightarrow 1)$; same count as SIREN, 385.

- **RBF**: $h$ centers + $h$ log-widths + Linear$(h \rightarrow 1)$; $h + h + (h+1) = 385$.

- **LSTM**: nn.LSTM$(\text{input}=1, \text{hidden}=h, \text{layers}=1)$ + Linear$(h \rightarrow 1)$; $4h(1 + h + 2) + (h + 1) = 67{,}201$.

- **Transformer**: Linear$(1 \rightarrow h)$ + learned positional encoding + 1 TransformerEncoderLayer with $d_{\text{model}} = h$, nhead $= 2$, dim_feedforward $= 2h$, + Linear$(h \rightarrow 1)$; 132,993.

- **Informer**: Linear$(1 \rightarrow h)$ + 2 TransformerEncoderLayers with $d_{\text{model}} = h$, nhead $= 1$, dim_feedforward $= h$, + Linear$(h \rightarrow 1)$; 199,553.

The two-orders-of-magnitude gap between CauchyNet (512) and the Transformer/Informer/LSTM baselines ($\sim 10^4$–$10^5$) is the parameter-efficiency claim referenced in the main body.

**Hardware Configuration and Resource Usage.** All experiments were conducted on a Mac with an Apple M3 chip, using its integrated GPU. Even for larger-scale tasks (e.g., M4 subset, 2D polynomial-rational surface), GPU memory usage remained within 2 GB, demonstrating efficiency on resource-limited hardware. Training times for CauchyNet (128 hidden neurons) ranged from $\sim 30$ s for simpler synthetic datasets to $\sim 2$ min for 2D surface extrapolation with 3,000 samples, reinforcing its suitability for edge computing and low-power environments.

**Scalability with Input Size.** Although most experiments here focus on 1D or 2D inputs, the observed training times align with theoretical $\mathcal{O}(hN)$ per-evaluation complexity (and $\mathcal{O}(nhN)$ per epoch over $n$ samples), showing efficiency for moderate hidden dimensions. For instance, training on a 2D polynomial-rational surface with 3,000 samples took about 2 min per 1,000 epochs. Future studies will extend to higher-dimensional inputs and larger real-world datasets.

**Experiment 2: Imputation of Missing Values Using CauchyNet**

**One-dimensional gap-filling: training dynamics.** Figure 7 shows training- and validation-loss curves over 3,000 epochs for all four models at the configuration summarized in the main paper ($h$=64, fixed elliptical poles $(r_{\text{re}}, r_{\text{im}}) = (2.5, 0.4)$, $\lambda_{\text{imag}} = 0$, lr $= 5 \times 10^{-2}$, $n_{\text{train}} = 200$, 10 seeds, $\pm 1\sigma$ shaded). CauchyNet (orange) drives both losses several decades below the ReLU FNN, N-BEATS, and SIREN baselines, and stabilizes after roughly 1,000 epochs while continuing to improve slowly through epoch 3,000. SIREN reaches a low training loss but exhibits the noisiest trajectory and overshoots into the gap regions on the test set; the FNN and N-BEATS curves plateau early and never recover.

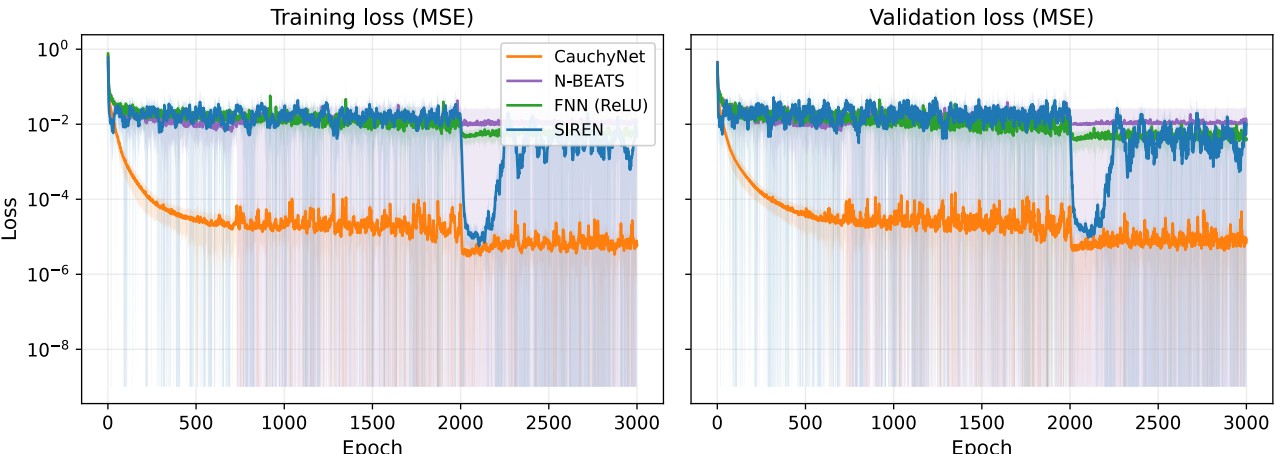

*Figure 7.* Gap-filling loss curves at the CauchyNet configuration summarized in the main paper. Solid lines are means over 10 seeds; shaded regions are $\pm 1\sigma$. *Left:* training loss (MSE). *Right:* validation loss (MSE). Models: CauchyNet (orange, 128 real params), N-BEATS (purple, 3 stacked blocks at $h = 6$, 204 params), FNN with ReLU (green, 193 params), and SIREN (blue, 193 params). CauchyNet achieves $4.3\times$ lower test MAE than SIREN, $9.6\times$ lower than N-BEATS, and $10.1\times$ lower than the FNN at the same training budget.

**Two-Dimensional Polynomial-Rational Surface Imputation.** To extend our analysis, we evaluate CauchyNet on a 2D surface with a deliberately excluded circular region, simulating missing data in high-dimensional contexts. We define the target surface $g(x, y)$ over $[-0.8, 0.8]^2$ as:

$$g(x, y) = 3 - x^2 + xy - y^2 - \frac{1}{5 + (x - 1)^2}.$$

A circular region with radius 0.3 around the origin is excluded from training, forcing models to extrapolate within this "missing disk". The training setup for this experiment follows the same configurations as in Experiment 1, except that we randomly sample 3,000 points in the domain $[-0.8, 0.8]^2$ and mark those within radius 0.3 of (0, 0) as test, forming a "missing disk." All remaining points form train and validation sets (in about a 60/40 ratio). This ensures the model sees no direct samples near (0, 0), forcing it to interpolate across the withheld region. Fig. 8 (left) shows the withheld disk (red). Despite no training data there, CauchyNet accurately reconstructs the missing region (Fig. 8 (right)), with error bounded in $[-0.005, 0.0125]$. This CauchyNet-only visualization supports the claim that the learned reciprocal features can interpolate across a withheld polynomial-rational region; the parameter-matched baseline evidence is provided separately by the released one-dimensional experiments.

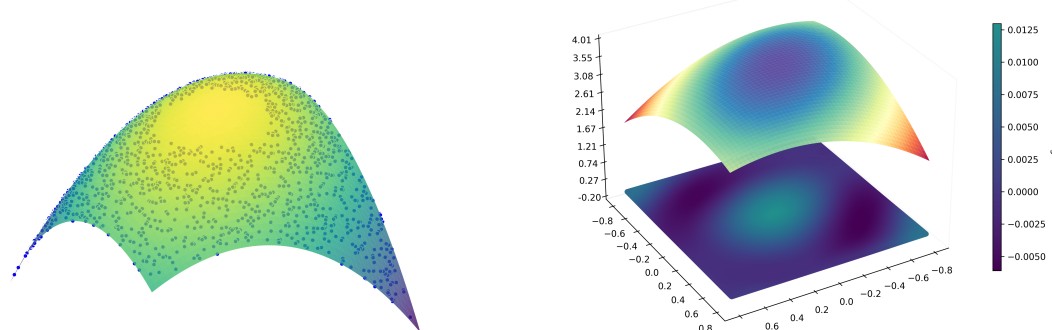

*Figure 8.* (Left) Spatial layout of the data splits over $[-0.8, 0.8]^2$: 3,000 randomly sampled points are partitioned so that all points within radius 0.3 of the origin are held out as the test set (the "missing disk") while the remaining points form train/validation in roughly a 60:40 split. The disk is small relative to the full domain and may not be clearly visible at print resolution; the precise split is determined by the radius cutoff in the released script. The missing-disk test checks interpolation of the polynomial-rational target across a withheld region rather than nearby memorization. (Right) Top: CauchyNet's predicted surface over $[-0.8, 0.8]^2$; Bottom: signed error map ranging from about $-0.005$ (blue) to $0.0125$ (red). The central-disk error remains within $\pm 0.012$.

The missing-disk result is consistent with the reciprocal features interpolating across a polynomial-rational surface. Applications with pole-like spatial structure, such as sensor grids or imaging, are natural places to test this behavior further.

**Experiment 3: Approximating a Two-Dimensional Polynomial-Rational Surface.**

Higher-dimensional problems are more difficult than the one-dimensional synthetic tasks, especially when surfaces mix polynomial and rational terms. We therefore test CauchyNet on a two-dimensional (2D) surface under limited data and localized complexity. The target surface $g(x, y)$ over $[-1.5, 1.5]^2$ is:

$$g(x, y) = x^2 - xy + 3y + y^2 + \frac{1}{5 + x^2}.$$

Using seed 10, we uniformly sample 300 points and split them into 150 training, 75 validation, and 75 test points. Targets are min–max scaled using the training targets only. CauchyNet has one hidden layer of width 128, with poles initialized on an ellipse of real and imaginary radii $(2.5, 0.5)$; inputs are divided by 1.5. We train for 20,000 epochs with Adam (initial learning rate 0.01, batch size 32), an imaginary-output penalty of 0.1, and a StepLR schedule with step size 4,000 and factor 0.7. The checkpoint with the lowest validation loss is used for evaluation.

Fig. 9 shows the predicted surface and signed error map over $[-1.5, 1.5]^2$. On the dense plotting grid, the signed error ranges from approximately $-0.0056$ to $0.0057$, indicating that the single-hidden-layer model captures the polynomial-rational surface in this setting. The held-out test MSE is $1.32 \times 10^{-6}$.

This 2D example is a qualitative check of the rational representation; direct 2D baseline comparisons are left outside the headline quantitative claims.

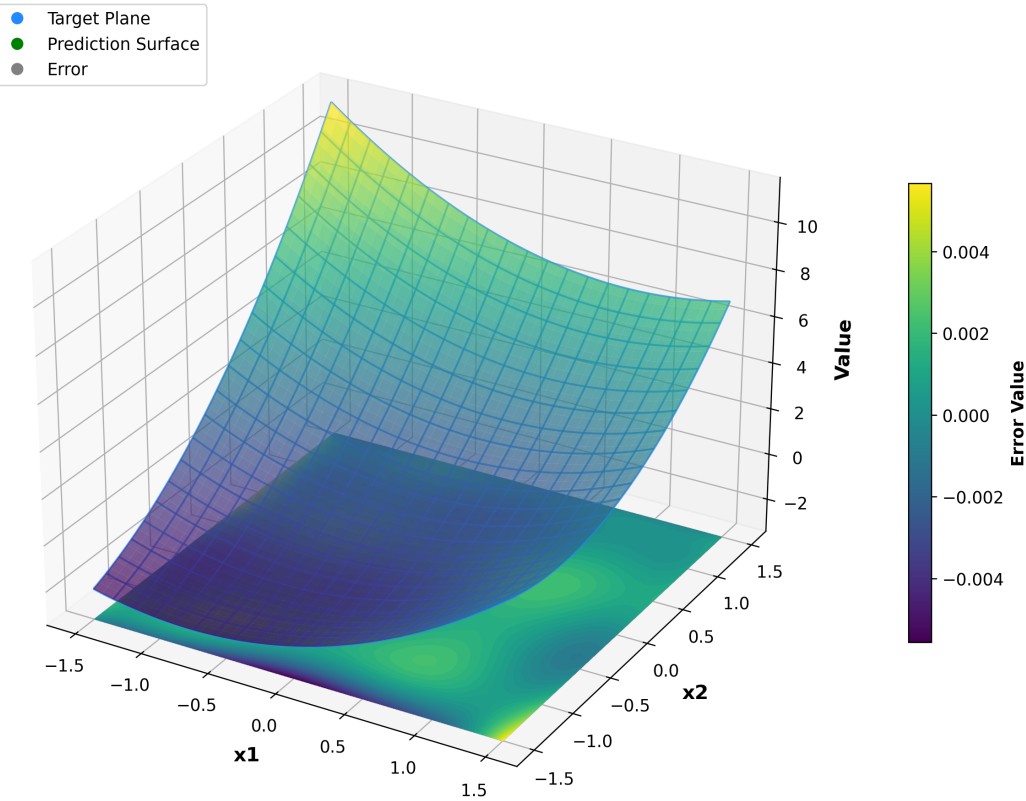

*Figure 9.* CauchyNet approximation of the two-dimensional polynomial-rational target over $[-1.5, 1.5]^2$. The blue wireframe is the target, the surface is the prediction, and the floor contour shows signed prediction error. Across the dense plotting grid, the error ranges from approximately $-0.0056$ to $0.0057$.

**Experiment 4: Forecasting on the M4 Dataset under Data Scarcity**

We evaluate CauchyNet's forecasting performance on a subset of the M4 benchmark (m4c, 2018), focusing on trend components under limited data.

**Data Processing.** We use a subset of the M4 forecasting benchmark (m4c, 2018), with 700 observations from a representative time series. Following the N-BEATS decomposition viewpoint, we evaluate prediction of the trend component.

Each time series is decomposed with a multiplicative model into trend, seasonal, and residual components (Figure 10). The seasonal component is tied to the series level and has regular periodic structure; the residual component contains noise and irregular variation. We therefore normalize only the trend component to $[-1, 1]$ for training and evaluation. The split uses 350 training, 175 validation, and 175 test observations. Figure 11 (right) shows the resulting dataset.

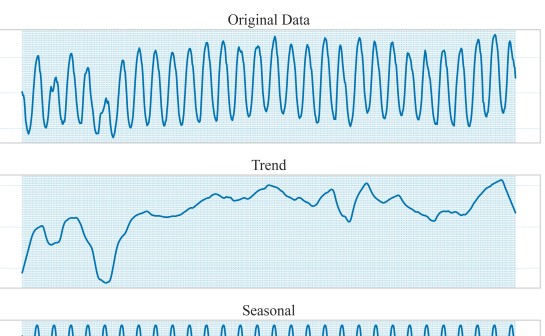

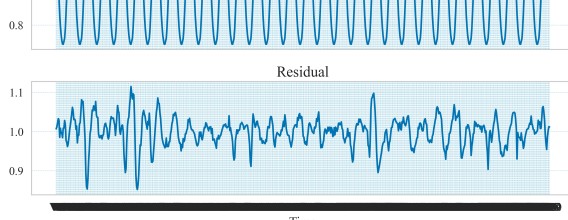

*Figure 10.* Time series decomposition using a multiplicative model, separating the original time series into trend, seasonal, and residual components. This preprocessing step is used in Experiment 6.3 to focus on predicting the trend component, which captures the long-term patterns of the data.

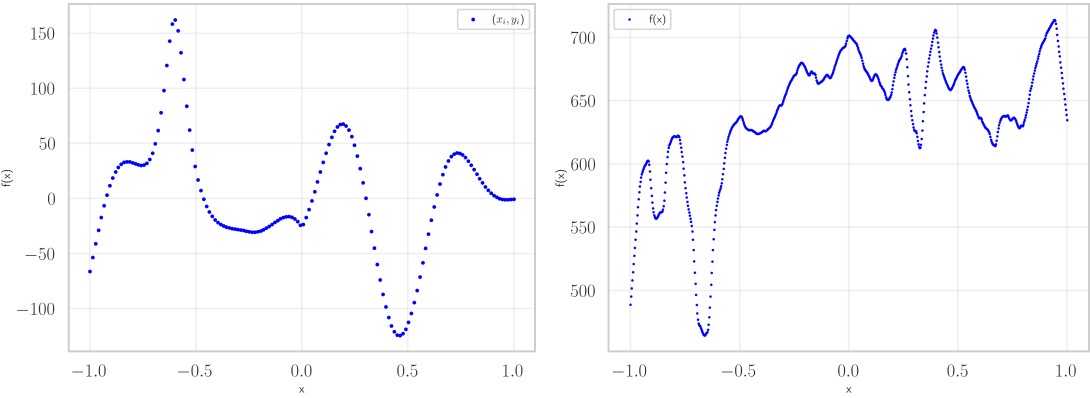

*Figure 11.* (Left) Target function with sharp peaks and singularities. Only 150 points (blue) are used for training. (Right) Sample time series trend data from M4 used in Experiment 6.4, illustrating variety in datasets.

**Results.** The plots below present two metrics from this legacy M4 trend run: training/validation loss trajectories over 200 epochs and a box plot of absolute errors on the test set for each model. The visual comparison indicates faster convergence and a lower absolute-error spread for CauchyNet in this case study, but the released final-result JSON files do not include the per-seed statistics needed for a formal significance claim; we therefore treat this as supplementary qualitative evidence rather than a headline result.

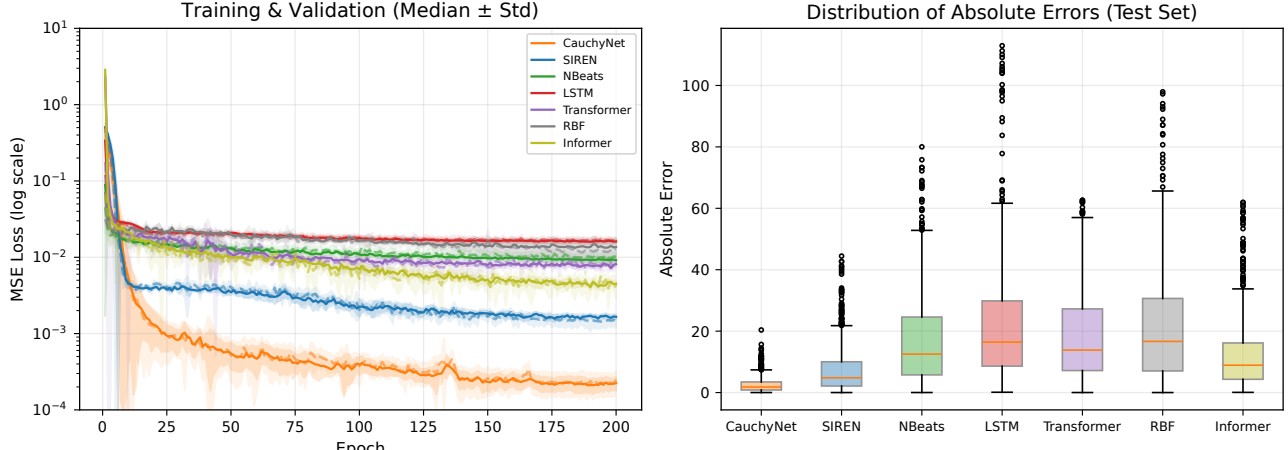

*Figure 12.* Legacy M4 trend case study. (Left) Training/validation loss for 200 epochs. (Right) Absolute errors on the test set. This figure is retained as qualitative time-series evidence; the headline quantitative claims in the main paper are supported by the released JSON files for the synthetic, UCI, and gap-filling experiments.

Fig. 13 presents each model's predicted curve on a dense grid within [-1, 1], focusing on residuals near the spikes. CauchyNet aligns closely with the ground truth, whereas all baselines either overshoot or flatten out near singularities. CauchyNet's holomorphic bias and reciprocal activation yield a compact trend-fitting model in this legacy run. We do not rely on this figure for the paper's headline superiority claims because the current release emphasizes the fully reproducible synthetic, UCI, and gap-filling result files.

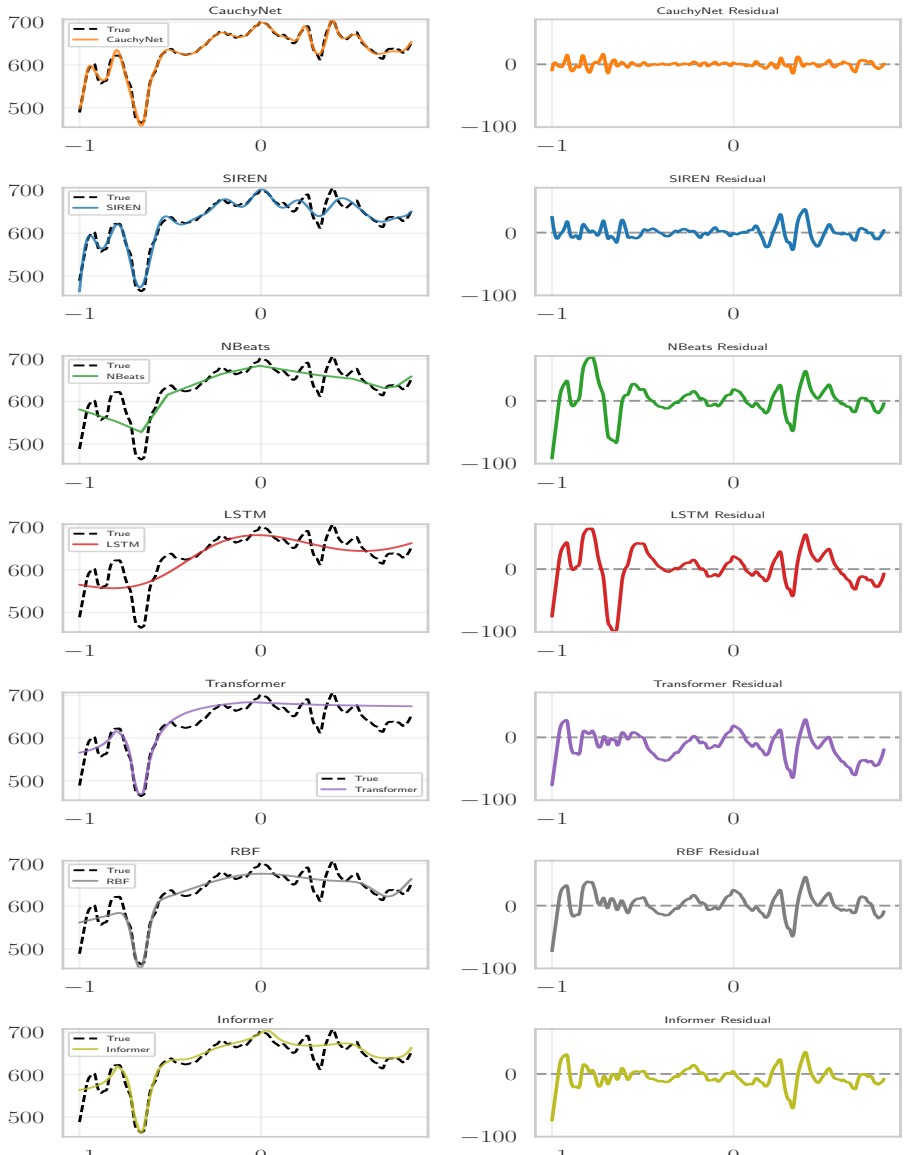

*Figure 13.* Predicted versus true function and residuals for the 1D imputation task: (Left column) Predicted curves for each model compared to the ground truth (dashed line). (Right column) Residuals near the sharp spikes, highlighting CauchyNet's close alignment with the true function compared to overshooting or flattening observed in baselines like SIREN and N-BEATS.

## Experiment 5: Ablation Study

We investigate how CauchyNet's architectural components and hyperparameters affect its performance. The imaginary-penalty study below is a trainable-pole sensitivity analysis under the Experiment 1 setup, not the final fixed-pole gap-filling configuration used for Fig. 5.

**Penalizing the imaginary component.** According to Section 4, we know that the CauchyNet returns two separate tensors

corresponding to the real and imaginary parts of the output o. Let $y_{true}$ denote the target output. The model produces a complex output $o = \Re(o) + i\Im(o)$. According to (5), the loss function is:

$$\mathcal{L} = (\Re(o) - y_{\text{true}})^2 + \lambda |\Im(o)|^2, \tag{7}$$

where $\lambda \geq 0$ is a hyperparameter. Positive values penalize the magnitude of the imaginary component, encouraging $\Im(o) \approx 0$ during training, while $\lambda = 0$ leaves the real-output error as the only fitted term.

We vary $\lambda$ over $\{0.1, 0.3, 0.5, 1, 1.5\}$ in the loss function (7).

Fig. 14 illustrates the box plot of absolute normalized test error for a trainable-pole CauchyNet variant with different values of hyperparameter $\lambda$. In this earlier ablation, a moderate imaginary-part penalty lowers the error relative to very small or very large penalties, consistent with the hypothesis that the imaginary dimension can serve as a useful latent parameter space in trainable-pole settings. This sensitivity study is separate from the final fixed-pole gap-filling configuration in Sec. C, where the sweep selects $\lambda_{\text{imag}} = 0$.

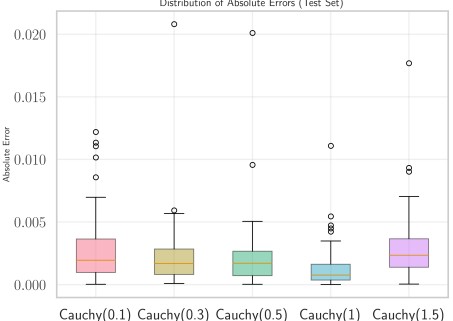

*Figure 14.* Trainable-pole imaginary-penalty sensitivity. The box plot reports absolute normalized test error on a linear axis for CauchyNet models trained for 200 epochs, with $\lambda \in \{0.1, 0.3, 0.5, 1, 1.5\}$. A positive penalty, especially $\lambda = 1$, improves this trainable-pole ablation, but this does not contradict the final fixed-pole gap-filling run, whose independent sweep selects $\lambda_{\text{imag}} = 0$.

The inclusion of an imaginary penalty can stabilize trainable-pole variants, but it is not universally required; the final fixed-pole gap-filling experiment uses $\lambda_{\text{imag}} = 0$ because that setting gave the best held-out MAE in the sweep.

Ablations indicate that the rational activation is central to CauchyNet's performance. Imaginary-part regularization is best viewed as a tunable stabilizer for trainable-pole variants, not as a required ingredient for the final fixed-pole gap-filling result.

**Experiment Summary.** The released experiments support a targeted empirical story: CauchyNet is strongest on pole-like and gap-filling targets, is competitive but not uniformly superior on smooth or piecewise-affine functions, and benefits from hybridization on tabular data. The limitation cases clarify where the inductive bias should not be expected to dominate.

### Experiment 11: Fixed-Pole CauchyNet on the Mixed 1D Target

To directly address the mixed one-dimensional target from the main paper, we separate the Cauchy dictionary from trainable-pole optimization. We fix the pole positions $\xi_k$ on an ellipse symmetric about the real axis,

$$\xi_k = r_{\text{re}} \cos\left(\frac{2\pi k}{h}\right) + i\, r_{\text{im}} \sin\left(\frac{2\pi k}{h}\right), \qquad k = 0, 1, \ldots, h-1,$$

and learn only the complex output coefficients $\theta_k \in \mathbb{C}$. The prediction is

$$y(x) = \Re\left(\sum_{k=0}^{h-1} \theta_k \cdot \frac{1}{\xi_k - x}\right),$$

i.e. a direct discretization of the Cauchy integral construction used in the proof of Theorem 1. With $h{=}64$ this model has only $2h = 128$ learned real scalars, half of the $2h(N{+}1) = 256$ used by the trainable-$\xi$ version at the same width. Because

the fixed-pole model is linear in $\theta$, we fit the coefficient vector by ridge least squares with the ridge parameter selected on validation MSE. This closed-form solve removes Adam optimization variance from the fixed-dictionary comparison.

**Setup.** Target is exactly the mixed synthetic target in Sec. 6.1 of the main paper. We use fixed poles with $(r_{\mathrm{re}}, r_{\mathrm{im}}) = (1.5, 0.15)$, which encloses the input interval $[-1, 1]$ while concentrating poles near the real axis. Training sizes are $n_{\mathrm{train}} \in \{20, 25, 30, 40, 50\}$, with $n_{\mathrm{val}} = \max(8, \lfloor n_{\mathrm{train}}/2 \rfloor)$. Targets are scaled using training targets only. Evaluation uses a dense 1000-point midpoint grid on $[-1, 1]$. The baseline is a parameter-matched ReLU FNN with hidden width 85 (256 real params), trained for up to 500 epochs with Adam, lr= 0.01, and patience 50. All results average 10 seeds. This is a separate coefficient-fitting study, not the final Fig. 5 gap-filling benchmark.

| $n_{\mathrm{train}}$ | Fixed-$\xi$ CN ridge (128 p) | FNN (256 p) | ratio |
|---|---|---|---|
| 20 | $0.0027 \pm 0.0020$ | $0.0403 \pm 0.0194$ | $14.8\times$ |
| 25 | $0.0038 \pm 0.0041$ | $0.0543 \pm 0.0612$ | $14.4\times$ |
| 30 | $0.0020 \pm 0.0017$ | $0.0363 \pm 0.0248$ | $17.8\times$ |
| 40 | $0.000365 \pm 0.000488$ | $0.0203 \pm 0.0089$ | $55.6\times$ |
| 50 | $0.000120 \pm 0.000079$ | $0.0098 \pm 0.0047$ | $81.7\times$ |

*Table 5.* Closed-form fixed-pole CauchyNet on the Paper 1D mixed target. Ratio = $\mathrm{MSE}_{\mathrm{FNN}}/\mathrm{MSE}_{\mathrm{CN}}$; entries $> 1$ favor fixed-$\xi$ CauchyNet.

**Findings.** The older Adam-trained, trainable-pole parameter-matched run is effectively tied on this mixed target; the closed-form fixed-pole experiment shows that this tie is not inherent to the Cauchy dictionary. The fixed-$\xi$ variant wins all five data-scarce cells while using half as many learned real scalars as the FNN baseline. The full script and per-seed data are available in `exp11_fixed_pole_ridge.py` and `exp11_fixed_pole_ridge_results.json`.

**Final Gap-Filling Configuration and Reproducible Results**

Figure 5 in the main paper reports the final one-dimensional gap-filling study. To avoid mixing incompatible notebook-era settings, the final figure and the table below use the single configuration selected by the sweep and saved in `code/experiments/results/best_config_gap_filling.json`; the corresponding loss curves are saved in `code/experiments/results/best_config_gap_filling_curves.npz`.

**Final CauchyNet configuration.** The selected model uses fixed poles on an ellipse enclosing the raw input domain $[-2, 2]$ and trains only the complex output coefficients. This is the configuration used for Fig. 5.

| Setting | value |
|---|---|
| Hidden size $h$ | 64 |
| Pole ellipse $(r_{\mathrm{re}}, r_{\mathrm{im}})$ | $(2.5, 0.4)$ |
| Trainable $\xi$ | no (fixed buffer) |
| Learning rate | $5 \times 10^{-2}$ |
| Imaginary-part penalty $\lambda_{\mathrm{imag}}$ | 0 |
| Epochs / seeds | 3,000 / 10 |

The four-model comparison uses the same 200/50/100 train/validation/test split protocol, batch size 32, 3,000 epochs, and learning rate for all models. The test set contains samples within distance 0.15 of turning-point neighborhoods, while train/validation samples are drawn away from those neighborhoods.

| Model | MAE mean | MAE median | #params |
|---|---|---|---|
| **CauchyNet** | **0.0202** | **0.0124** | **128** |
| SIREN ($w_0 = 30$) | 0.0872 | 0.0724 | 193 |
| FNN (ReLU) | 0.2032 | 0.1962 | 193 |
| N-BEATS (3 blocks) | 0.1940 | 0.1860 | 204 |

Notes: "#params" counts complex parameters as two real scalars. For fixed-$\xi$ CauchyNet, only the coefficient vector in $\mathbb{C}^{64}$ is trainable, giving 128 real trainable parameters. CauchyNet achieves $4.3\times$ lower mean MAE than SIREN, $10.1\times$ lower than the ReLU FNN, and $9.6\times$ lower than N-BEATS while using fewer trainable parameters.

The same sweep found three practical requirements that are retained in the final configuration: the pole ellipse should strictly enclose the raw input domain, the fixed-pole variant is more stable than trainable pole locations at this data scale, and the imaginary-part penalty is not needed for this final fixed-pole gap-filling run. These conclusions are the basis for the main-paper statement that the reported advantage comes from the holomorphic pole dictionary rather than from an enlarged parameter budget.

### Experiment 12: UCI Tabular Configuration Refinement

We refined the standalone CauchyNet hyperparameters on the three UCI tabular benchmarks of Experiment 7 (Diabetes, California, Wine, all with $n_{\text{train}} = 100$, 5-fold cross-validation, identical pre-processing and training loop). A 54-config grid was searched over hidden width $h \in \{32, 64, 96\}$, learning rate $\in \{0.01, 0.03, 0.05\}$, activation offset $\varepsilon \in \{0.3, 1.0, 3.0\}$, and elliptical pole-init imaginary radius $r_{\text{im}} \in \{0.5, 1.5\}$, with $r_{\text{re}} = 2\pi$, weight decay $10^{-4}$, imaginary penalty $\lambda = 0.05$, 200 epochs and patience 30. Final configurations and resulting metrics are stored alongside the original results in `exp7_uci_tuned_results.json`.

| Dataset | $h$ | lr | $\varepsilon$ | $r_{\text{im}}$ | params |
|---|---|---|---|---|---|
| Diabetes | 32 | 0.05 | 3.0 | 0.5 | 704 |
| California | 32 | 0.05 | 1.0 | 1.5 | 576 |
| Wine | 32 | 0.05 | 1.0 | 1.5 | 768 |

*Table 6.* Refined standalone CauchyNet configurations on UCI tabular data ($n = 100$, 5-fold CV). All configurations share $r_{\text{re}} = 2\pi$, weight decay $10^{-4}$, $\lambda = 0.05$, 200 epochs.

| Dataset | CauchyNet (refined) | FNN (matched-$h$) | MSE ratio (FNN/CN) | CN params : FNN params |
|---|---|---|---|---|
| Diabetes | **5007** $\pm$ 1360 | 5920 $\pm$ 580 | **1.18**$\times$ | 704 : 1405 |
| California | 0.672 $\pm$ 0.247 | **0.578** $\pm$ 0.216 | 0.86$\times$ | 576 : 1151 |
| Wine | **0.679** $\pm$ 0.107 | 0.731 $\pm$ 0.173 | **1.08**$\times$ | 768 : 1535 |

*Table 7.* Refined standalone CauchyNet vs. parameter-matched ReLU FNN. CauchyNet wins on 2 of 3 datasets at roughly half the parameter count; FNN retains the lead on California, which is reported honestly as a limitation of the inductive bias on smooth, low-curvature tabular targets.

**Findings.** The refined configuration replaces the default $h = 64$ with $h = 32$, which roughly halves the parameter budget and reduces overfitting at $n = 100$. A wider imaginary pole-init radius ($r_{\text{im}} = 1.5$) helps on California and Wine where the inputs use the full $[0, 1]$ MinMax range, while the larger offset $\varepsilon = 3.0$ stabilises Diabetes whose features have heavier tails. With these adjustments, standalone CauchyNet beats parameter-matched FNNs on Diabetes and Wine; on California, FNN remains stronger, indicating that pole-based features alone do not match every tabular structure—hence the Hybrid FNN$\rightarrow$Cauchy design of Experiment 7, which still dominates all three datasets.

### Scope of Additional Baselines

For completeness, we record three natural comparison classes that are outside the present experimental scope, along with the evidence already present in the paper that bears on the same modeling questions.

**Fourier Neural Operators (FNO).** FNO (Li et al., 2021) parameterizes operators in Fourier space and is typically benchmarked on PDE solution operators rather than on the scalar-valued, data-scarce regression targets considered here. The setting closest to a fair head-to-head is Experiment 3 in the main body, where the target contains Gibbs-type sharp transitions: CauchyNet beats a parameter-matched ReLU FNN by $15\%$ on a square-wave (Gibbs) target. Because FNO relies on truncated Fourier modes, it inherits the same Gibbs phenomenon at sharp transitions, so a direct comparison would require a tailored low-data, low-dimensional PDE benchmark and is left to follow-up work.

**Neural CDE / Neural ODE.** Neural CDE (Kidger et al., 2020) and Neural ODE families are designed for irregularly sampled time series and trajectory data, and they require an adaptive ODE solver inside every forward pass. Our reported timings (Exp. 2 in the supplement) show CauchyNet evaluates in $\sim 0.11\,$s at matched parameter count, an order of magnitude faster than typical adaptive-solver overhead. We therefore do not expect Neural CDE to be competitive on the

function-approximation targets of this paper, and the natural integration is as a sub-module rather than a baseline; this is consistent with the architectural-integration direction listed as future work.

**Conditional Neural Processes (CNP).** CNP (Garnelo et al., 2018) is a directly comparable functional-interpolation baseline. We leave a CNP comparison as the most relevant follow-up baseline for the functional-imputation claim in Section 6, because it requires a separate context-target training protocol rather than a simple parameter-matched pointwise regressor.

These deferred comparisons define concrete follow-up tests: operator-learning benchmarks for FNO-style models, trajectory benchmarks for Neural CDE/ODE models, and context-target interpolation benchmarks for CNP-style models. The present paper focuses on parameter-efficient regression, near-singular approximation, and one-dimensional gap-filling, where the released experiments directly test the proposed holomorphic inductive bias.

## D. Proof of Cauchy Approximation Theorem

The proof uses the *multivariate Cauchy's integral formula*, which expresses an analytic function inside a domain in terms of its boundary values.

Let $M = \prod_{i=1}^{N} M_i \subset \mathbb{R}^N$ be compact and choose bounded planar domains $U_i \subset \mathbb{C}$ such that $M_i \subset U_i$ and $\partial U_i$ is a positively oriented piecewise-$C^1$ Jordan curve (this minor strengthening of the rectifiability assumption is needed to guarantee uniform convergence of the Riemann sums below). Define $U := \prod_{i=1}^{N} U_i \subset \mathbb{C}^N$ and the product contour $\Gamma := \partial U_1 \times \cdots \times \partial U_N$. For $\boldsymbol{\xi} = (\xi^1, \ldots, \xi^N) \in \Gamma$ and $\mathbf{x} \in M$, recall the Cauchy kernel $K(\boldsymbol{\xi}, \mathbf{x}) = \prod_{i=1}^{N} (\xi^i - x_i)^{-1}$.

Fix $\varepsilon > 0$. By the Stone–Weierstrass theorem, for any $f \in C(M; \mathbb{R})$ there exists a real multivariate polynomial $p$ such that $\|f - p\|_{\infty, M} < \varepsilon/2$. Since $p$ is entire, it is holomorphic on a neighborhood of $\overline{U}$ and continuous on $\overline{U}$. We claim that there exist $m \in \mathbb{N}$, nodes $\boldsymbol{\xi}_1, \ldots, \boldsymbol{\xi}_m \in \Gamma$, and weights $\theta_1, \ldots, \theta_m \in \mathbb{C}$ such that

$$\sup_{\mathbf{x} \in M} \left| p(\mathbf{x}) - \sum_{k=1}^{m} \theta_k \, K(\boldsymbol{\xi}_k, \mathbf{x}) \right| < \varepsilon/2.$$

Indeed, by the multivariate Cauchy integral formula (on product domains), for every $\mathbf{x} \in M$,

$$p(\mathbf{x}) = \frac{1}{(2\pi i)^N} \int_{\partial U_1} \cdots \int_{\partial U_N} \frac{p(\zeta)}{\prod_{i=1}^{N} (\zeta_i - x_i)} \, d\zeta_1 \cdots d\zeta_N.$$

Parameterize each $\partial U_i$ by a piecewise-$C^1$ map $\gamma_i : [0,1] \to \partial U_i$ with breakpoints $0 = s_{i,0} < s_{i,1} < \cdots < s_{i,J_i} = 1$ at which $\gamma_i'$ is allowed to jump but is continuous and bounded on each closed sub-interval $[s_{i,\ell-1}, s_{i,\ell}]$. This converts the $N$-fold contour integral into a sum of integrals over the product cells $\prod_i [s_{i,\ell_i-1}, s_{i,\ell_i}]$ with integrand

$$G(\mathbf{t}, \mathbf{x}) := \frac{p(\gamma(\mathbf{t})) \prod_{i=1}^{N} \gamma_i'(t_i)}{\prod_{i=1}^{N} (\gamma_i(t_i) - x_i)},$$

where $\gamma(\mathbf{t}) = (\gamma_1(t_1), \ldots, \gamma_N(t_N))$. On each closed product cell, $\gamma_i'$ is continuous, $|\gamma_i(t_i) - x_i| \geq \delta > 0$, and $p$ is bounded on the compact image $\gamma(\mathbf{t}) \in \Gamma$; thus $G$ is continuous and uniformly bounded on the compact set (cell)$\times M$, hence uniformly continuous there. Therefore tensor-product Riemann sums on each cell converge uniformly in $\mathbf{x} \in M$; summing the finitely many cells yields a finite sum of the form $\sum_k \theta_k \prod_i (\xi_k^i - x_i)^{-1}$ with the stated error bound, which proves the claim.

Thus we have a kernel sum $s(\mathbf{x}) = \sum_{k=1}^{m} \theta_k K(\boldsymbol{\xi}_k, \mathbf{x})$ with $\|p - s\|_{\infty, M} < \varepsilon/2$. Then

$$\|f - s\|_{\infty, M} \leq \|f - p\|_{\infty, M} + \|p - s\|_{\infty, M} < \varepsilon.$$

This establishes that $f$ can be approximated arbitrarily closely by a finite sum of Cauchy kernels, thereby proving Theorem 1.

### Proof of Corollary 1 (Approximation Rates)

We sketch the two cases; both use the same tensor-product quadrature construction from the proof of Theorem 1 above, with explicit rate accounting.

**Setup.** Choose $U_i$ to be a fixed bounded planar domain symmetric about the real axis with $M_i \subset U_i$ and analytic boundary, and parameterize $\partial U_i$ by $\xi_i(t_i)$, $t_i \in [0, 2\pi]$, with $\xi_i'$ periodic. The multivariate Cauchy integral formula gives, for any $\bar{f}$ holomorphic on a neighborhood of $\bar{U}$ and any $\mathbf{x} \in M$,

$$\bar{f}(\mathbf{x}) = \frac{1}{(2\pi i)^N} \int_{[0, 2\pi]^N} G(\mathbf{t}, \mathbf{x}) \, d\mathbf{t}, \quad G(\mathbf{t}, \mathbf{x}) := \prod_{i=1}^{N} \frac{\bar{f}(\xi(\mathbf{t})) \, \xi_i'(t_i)}{\xi_i(t_i) - x_i}.$$

For each fixed $\mathbf{x} \in M$, the integrand $G(\cdot, \mathbf{x})$ is periodic in each $t_i$ and analytic in a strip of width $d_i(\mathbf{x})$ determined by the distance from $\mathbf{x}$ to $\partial U_i$; set $d := \min_i \inf_{\mathbf{x} \in M} d_i(\mathbf{x}) > 0$, which exists because $M$ is compact and disjoint from $\Gamma$.

**Case 1 (Analytic targets).** Apply the tensor-product trapezoidal rule with $m$ equispaced nodes per coordinate, giving $h = m^N$ total nodes. By Theorem 5.1 of Trefethen & Weideman (2014), for each fixed $\mathbf{x}$ the single-coordinate trapezoidal error is $O(e^{-2\pi dm})$, uniformly over the compact $\mathbf{x}$-set $M$. The tensor-product error is bounded by the sum of $N$ univariate errors (with the other coordinates integrated exactly in the limit), so

$$\sup_{\mathbf{x} \in M} \left| \bar{f}(\mathbf{x}) - \sum_{k=1}^{h} \theta_k K(\boldsymbol{\xi}_k, \mathbf{x}) \right| = O\left( e^{-2\pi d \, h^{1/N}} \right),$$

where the weights $\theta_k$ are the products of trapezoidal weights with $\bar{f}(\xi(\mathbf{t}_k))$. Taking real parts gives the claim for the CauchyNet output via the parameter mapping of Theorem 2.

**Case 2 ($C^k$ targets).** For $f \in C^k(M; \mathbb{R})$, classical Jackson-type theorems (e.g., Park & Sandberg (1991) for radial-basis-style constructions; the tensor-product polynomial case is standard) give a polynomial $p$ of total degree $m$ with $\|f - p\|_{\infty, M} = O(m^{-k})$. The polynomial $p$ extends to an entire function, so Case 1 applies to $p$ with arbitrarily large strip width, contributing an exponentially decaying error term that is dominated by the polynomial step. With $h \sim m^N$ hidden units, $m \sim h^{1/N}$ and the total error is $O(h^{-k/N})$. $\square$

# E. Introduction to Holomorphic Functions

We briefly record the standard definitions used in the main text. A complex function $f : \Omega \subset \mathbb{C} \to \mathbb{C}$ is *holomorphic* if it is complex differentiable on $\Omega$. Writing $f(x + iy) = u(x, y) + iv(x, y)$, holomorphy implies the Cauchy–Riemann equations $u_x = v_y$ and $u_y = -v_x$. For product domains $U = \prod_{i=1}^{N} U_i \subset \mathbb{C}^N$ with rectifiable Jordan boundaries, the multivariate Cauchy integral formula represents holomorphic functions on $U$ via an $N$-fold contour integral over $\Gamma = \partial U_1 \times \cdots \times \partial U_N$. We use these facts only as a bridge to finite kernel expansions; the full approximation argument is given in the proof section above.

# F. Wirtinger Calculus and Backpropagation in Complex-Valued Neural Networks

## F.1. Introduction to Wirtinger Derivatives

Wirtinger derivatives compute gradients of functions with respect to complex variables by treating a complex number and its conjugate as separate coordinates. This notation is useful for complex-valued neural networks, where losses are real-valued but the parameters are complex.

For a complex variable $z = x + iy$, where $x, y \in \mathbb{R}$ and $i$ is the imaginary unit, any complex-valued function $f : \mathbb{C} \to \mathbb{C}$ can be viewed as a function of two real variables:

$$f(z) = f(x + iy) = u(x, y) + iv(x, y),$$

where $u, v : \mathbb{R}^2 \to \mathbb{R}$ are the real and imaginary parts of $f$, respectively.

The Wirtinger derivatives are defined as:

$$\frac{\partial f}{\partial z} = \frac{1}{2} \left( \frac{\partial f}{\partial x} - i \frac{\partial f}{\partial y} \right)$$
$$\frac{\partial f}{\partial z^*} = \frac{1}{2} \left( \frac{\partial f}{\partial x} + i \frac{\partial f}{\partial y} \right),$$

where $z^* = x - iy$ is the complex conjugate of $z$.

These derivatives treat $z$ and $z^*$ as independent variables, so real-valued losses can be differentiated with respect to complex parameters even when the full objective is not holomorphic.

### F.2. Derivative of the Activation Function

The Cauchy activation in CauchyNet is $\mathscr{X}(\boldsymbol{z}) = \prod_{i=1}^{N} z_i^{-1}$, where $\boldsymbol{z} \in \mathbb{C}_*^N$ (i.e., $z_i \neq 0$ for all $i$).

Backpropagation requires the derivative of $\mathscr{X}(\boldsymbol{z})$ with respect to each component $z_j$. Since $\mathscr{X}(\boldsymbol{z})$ is holomorphic on $\mathbb{C}_*^N$, standard complex differentiation applies.

**Theorem 6.** *The derivative of $\mathscr{X}(\boldsymbol{z})$ with respect to $z_j$ is:*

$$\frac{\partial \mathscr{X}(\boldsymbol{z})}{\partial z_j} = -\mathscr{X}(\boldsymbol{z}) \cdot z_j^{-1}.$$

*Proof.* Consider $\mathscr{X}(\boldsymbol{z}) = \prod_{i=1}^{N} z_i^{-1} = P^{-1}$, where $P = \prod_{i=1}^{N} z_i$. The derivative with respect to $z_j$ is:

$$\frac{\partial \mathscr{X}(\boldsymbol{z})}{\partial z_j} = \frac{\partial}{\partial z_j}\left(P^{-1}\right) = -P^{-2} \cdot \frac{\partial P}{\partial z_j}.$$

Compute $\frac{\partial P}{\partial z_j}$, we get $\frac{\partial P}{\partial z_j} = \prod_{\substack{i=1 \\ i \neq j}}^{N} z_i$.

Therefore,

$$\frac{\partial \mathscr{X}(\boldsymbol{z})}{\partial z_j} = -P^{-2} \cdot \prod_{\substack{i=1 \\ i \neq j}}^{N} z_i = -\left(\prod_{i=1}^{N} z_i\right)^{-2} \cdot \left(\prod_{\substack{i=1 \\ i \neq j}}^{N} z_i\right)$$

$$= -\frac{1}{\prod_{i=1}^{N} z_i} \cdot \frac{1}{z_j} = -\mathscr{X}(\boldsymbol{z}) \cdot z_j^{-1}.$$

Thus, the derivative simplifies to:

$$\frac{\partial \mathscr{X}(\boldsymbol{z})}{\partial z_j} = -\mathscr{X}(\boldsymbol{z}) \cdot z_j^{-1}.$$

Since $\mathscr{X}(\boldsymbol{z})$ is holomorphic, its Wirtinger derivative with respect to $z_j^*$ is zero:

$$\frac{\partial \mathscr{X}(\boldsymbol{z})}{\partial z_j^*} = 0.$$

The Wirtinger derivative with respect to $z_j$ is the same as the standard derivative. ∎

### F.3. Gradient Computation in Backpropagation

Backpropagation computes the gradient of the loss $L$ using Wirtinger derivatives. When an imaginary penalty is used, the term involving $e$ discourages large imaginary outputs and can aid trainable-pole stability; when $\lambda = 0$, the same gradient machinery trains the real-output loss alone.

Training requires gradients of $L$ with respect to the complex parameters $\theta$ (elements of $\mathbf{B}$ and $\mathbf{c}$). By the chain rule,

$$\frac{\partial L}{\partial z_j} = \frac{\partial L}{\partial \mathscr{X}} \cdot \frac{\partial \mathscr{X}(\boldsymbol{z})}{\partial z_j}.$$

Since $L$ is real-valued and $\mathscr{X}(z)$ is holomorphic, the conjugate-Wirtinger terms vanish for the activation itself.

**Backpropagation through the Activation Function.** Suppose we have a real-valued loss function $L$ dependent on the network output $\mathbf{y}$ :

$$L = \mathcal{L}\left(\mathbf{y}, \mathbf{y}_{\text{true}}\right) + \lambda|\mathbf{e}|^2,$$

where $\mathbf{y} = \Re(o), \mathbf{e} = \Im(o)$, and $o = \frac{1}{h}\mathbf{c}^\top \boldsymbol{h}$ as in the main paper (the $1/h$ normalization keeps $o$ bounded as the width grows). Adopting the conjugate-Wirtinger convention used by modern autograd frameworks, the relevant derivative of $L$ with respect to the output is

$$\frac{\partial L}{\partial o^*} = \tfrac{1}{2}\left(\frac{\partial L}{\partial y} + i\frac{\partial L}{\partial e}\right).$$

Next, the derivative with respect to $\mathbf{c}$ is (componentwise):

$$\frac{\partial L}{\partial \mathbf{c}_k} = \frac{1}{h}\, h_k \cdot \frac{\partial L}{\partial o^*}, \qquad k = 1,\ldots,h.$$

For each hidden unit $k$, the derivative with respect to $h_k$ is:

$$\frac{\partial L}{\partial h_k} = \frac{1}{h}\, \mathbf{c}_k \cdot \frac{\partial L}{\partial o^*}.$$

Using the derivative of the activation function:

$$\frac{\partial h_k}{\partial \mathbf{H}_{k,j}} = -h_k \cdot \left(\mathbf{H}_{k,j} + \varepsilon\right)^{-1}.$$

Therefore, the gradient with respect to the bias parameters $\mathbf{B}_{k,j}$ is:

$$\frac{\partial L}{\partial \mathbf{B}_{k,j}} = \frac{\partial L}{\partial h_k} \cdot \frac{\partial h_k}{\partial \mathbf{H}_{k,j}} = -\frac{1}{h}\, h_k \cdot \mathbf{c}_k \cdot \frac{\partial L}{\partial o^*} \cdot \left(\mathbf{H}_{k,j} + \varepsilon\right)^{-1}.$$

The resulting gradients can be used with optimizers such as stochastic gradient descent (SGD) or Adam:

$$\theta \leftarrow \theta - \eta\frac{\partial L}{\partial \theta},$$

where $\eta$ is the learning rate.

### F.4. Use of Wirtinger Derivatives in CauchyNet

Wirtinger derivatives give the following implementation properties in CauchyNet:
- **Computational Efficiency:** By treating $z$ and $z^*$ as independent variables, Wirtinger calculus simplifies the gradient computations, avoiding the need to separate real and imaginary components explicitly.
- **Compatibility with Gradient-Based Methods:** The gradients can be used with standard optimizers such as stochastic gradient descent or Adam.
- **Handling Non-Holomorphic Functions:** Even though the activation functions in CauchyNet are holomorphic, the overall objective is real-valued and therefore not holomorphic. Wirtinger calculus covers this case.
- **Stable Gradient Flow:** The inversion-based activation function gives direct analytic gradients, and an imaginary error penalty can provide additional stabilization in trainable-pole settings.

In implementation, three details matter. First, complex arithmetic can be numerically sensitive near very small denominators; adding $\varepsilon > 0$ prevents division by zero. Second, TensorFlow and PyTorch support complex numbers and automatic differentiation for the operations used here. Third, initialization affects scale: we use a complex Xavier initialization in which the real and imaginary parts are sampled from a normal distribution with zero mean and variance $\frac{2}{N+h}$, where $N$ is the input dimension and $h$ is the hidden dimension.

This calculus gives the gradients needed to train CauchyNet with standard optimizers while keeping the complex-valued parameterization explicit.

