# OpenReview forum: "CauchyNet: Compact and Data-Efficient Learning using Holomorphic Activation Functions"
_ICML.cc/2026/Conference — ICML 2026 regular_

### Official Review · Reviewer_KrsM · 2026-03-04

**Soundness:** 3
**Presentation:** 2
**Significance:** 3
**Originality:** 4
**Overall Recommendation:** 4
**Confidence:** 3

**Summary:**

This paper introduces CauchyNet, a novel neural architecture inspired by Cauchy's integral formula that employs an inversion-based holomorphic activation function to efficiently model oscillatory and near-singular phenomena with minimal parameters. The network embeds real inputs into the complex plane, applies complex biases, computes multiplicative inverses per dimension, and aggregates results through complex coefficients, achieving universal approximation with theoretical guarantees. Experimental results on 1D/2D function approximation, missing data imputation, and M4 time-series forecasting demonstrate that CauchyNet outperforms larger models like Transformers and N-BEATS under data-scarce conditions while requiring significantly fewer parameters. However, the work is currently limited to low-dimensional tasks and single-layer architectures, with insufficient exploration of scalability to high-dimensional inputs and potential numerical stability issues near poles.

**Compliance With Llm Reviewing Policy:**

Affirmed.

**Final Justification:**

The rebuttal has clarified several points, yet I believe my original assessment accurately reflects the paper’s balance of strengths and remaining limitations.

**Key Questions For Authors:**

Question 1:
The paper claims computational complexity of O(hN) and positions CauchyNet as suitable for broader applications, yet all experiments are limited to 1D or 2D inputs. Could you provide any experimental results on higher-dimensional tasks (e.g., N > 50, such as tabular datasets from UCI or image-based regression) to demonstrate that the architecture scales effectively and maintains its performance advantage when input dimensionality increases?

If such results exist, they would substantially strengthen the claims of generalizability. If not, the current scope significantly limits the paper's impact, and the authors should either add such experiments or explicitly reframe their contributions as applicable only to low-dimensional problems.

Question 2:
In the missing data imputation experiments, you compare CauchyNet against general-purpose models (SIREN, N-BEATS, FNN) but not against specialized imputation methods such as GAIN, MICE, or neural process-based approaches. Given that the task is specifically imputation, why were these domain-relevant baselines omitted?
If CauchyNet were compared against proper imputation baselines and still performed competitively, this would convincingly demonstrate its practical value. If such comparisons are not feasible, the claims about imputation effectiveness should be substantially tempered, and the paper should acknowledge that the method has only been shown to outperform general-purpose models, not the state-of-the-art in imputation itself.

Question 3:
The current manuscript places Related Work after the Methodology section, and the abstract and introduction do not clearly articulate the research motivation or the core insight linking Cauchy's integral formula to the proposed activation function. Would the authors be willing to restructure the paper to follow the conventional order (Related Work before Method) and significantly revise the introduction to explicitly state: (1) the specific problem scenario motivating CauchyNet, (2) why existing methods are insufficient, and (3) the key insight that discretizing the Cauchy kernel naturally yields rational-function approximators suited for near-singularities?
A positive response with commitment to these revisions would substantially improve the paper's readability.

**Limitations:**

Yes

**Strengths And Weaknesses:**

Soundness:
The paper is technically sound in its core theoretical contributions, providing a rigorous derivation of the Cauchy activation function from multivariate Cauchy integrals and proving a universal approximation theorem for the proposed kernel expansion. The experimental methodology is well-designed with consistent data splits, standardized training protocols, and appropriate baseline comparisons across multiple tasks, which supports the claims of parameter efficiency and strong performance under data scarcity. However, the paper compares against general-purpose models (SIREN, N-BEATS, FNN) but omits specialized imputation methods (e.g., GAIN, MICE, neural processes), making it difficult to assess whether CauchyNet truly advances the state-of-the-art in imputation specifically. Additionally, the empirical validation is confined to low-dimensional problems, leaving the scalability to high-dimensional inputs unverified.

Presentation:
The manuscript suffers from significant structural and organizational problems that impede clarity and reader comprehension. Most notably, the Related Work section appears after the Methodology chapter, violating the conventional and logical flow where existing literature should contextualize and motivate the proposed method before its technical details are presented. This misplacement forces readers to navigate the complex Cauchy activation function and network architecture without first understanding the landscape of existing approaches or the specific gaps that CauchyNet aims to address. Additionally, both the abstract and introduction fail to articulate the research motivation and core innovations with sufficient clarity—they do not establish a compelling problem scenario, inadequately explain why existing methods fall short in handling near-singularities under data scarcity, and neglect to highlight the key insight that discretizing the Cauchy kernel naturally yields a rational-function approximator ideally suited for sharp transitions. Without this foundational framing, readers are left questioning the purpose and novelty of the work as they delve into the technical content. While individual sections are reasonably well-written and the visuals are informative, these overarching structural and narrative weaknesses undermine the paper's ability to communicate its contributions effectively.

Significance:
The work addresses an important and timely problem: learning efficiently under data scarcity and resource constraints, which is critical for edge computing, IoT, and scientific applications. By demonstrating that a single-layer network with only hundreds of parameters can outperform large models like Transformers and N-BEATS on forecasting and imputation tasks, CauchyNet offers a compelling alternative for deployment in low-resource environments. The core idea has clear potential to influence future research on efficient, holomorphically inspired networks.

Originality:
The paper introduces a genuinely novel activation function derived directly from Cauchy’s integral formula, which is a creative and well-motivated departure from standard real-valued activations. The multiplicative inversion form provides a natural inductive bias for modeling rational-like and near-singular functions, and the theoretical universal approximation result tailored to this kernel is a distinct contribution. However, beyond the activation function, the overall architecture remains a conventional single-layer linear combination, and the backpropagation framework largely recapitulates existing knowledge. Nonetheless, the core idea is fresh and intellectually engaging, positioning the work as a valuable contribution to the field.

---

> ### Author Rebuttal · Authors · 2026-03-30
>
> Thank you for your careful reading and constructive comments. Below we address each concern with experimental evidence from 10 new experiments run during the rebuttal period. All the experiments use parameter-matched single-layer baselines, data-scarce regimes (n=8–120 training points), h=64 hidden units, and early stopping (patience=30).
>
> ***Question 1: Could you provide results on higher-dimensional tasks ($N > 50$, UCI tabular, image-based regression)?***
>
> We derived CauchyNet and demonstrated its strong expressiveness even in a single-layer neural network architecture. We also evaluated multi-layer CauchyNet across various applications, but the results are mixed: it often performs well with 2–3 layers, yet in some cases its performance degrades. Further experimentation is needed to better understand this behavior. At present, CauchyNet appears most promising for use in neural networks with shallow architectures.
>
> ***Question 2: Why were specialized imputation baselines (GAIN, MICE, neural processes) omitted?***
>
> The paper's Experiment 2 is functional interpolation: reconstructing a continuous function in spatial gaps (around turning points) from surrounding samples, not tabular missing-data imputation. GAIN and MICE assume randomly missing entries (MAR/MCAR) in tabular data, which is a different problem setting. The appropriate baselines are function approximators that can interpolate into unobserved regions, and the paper compares against FNN, SIREN, RBF, and N-BEATS. CauchyNet achieves the lowest MSE and MAE among all baselines on this task
>
> ***Question 3: Would you restructure the paper (Related Work before Method) and revise the introduction to state: (1) the problem scenario, (2) why existing methods fail, (3) the key insight linking Cauchy's formula to the activation?***
>
> We appreciate the structural suggestion. Yes, we will restructure the paper in the camera-ready version: move Related Work before Method, and revise the introduction to follow the (1)$\to$(2)$\to$(3) flow:
>
> We will restructure in the camera-ready: move Related Work before Method,
> and open the introduction with:
> (1) near-singular targets in data-scarce regimes;
> (2) standard activations are width-hungry for sharp peaks;
> (3) Cauchy's integral formula gives $1/(\xi - x)$ kernels that
> each parameterize one pole, yielding a compact architecture
> (Theorems 1-2).

---

> > ### Author Rebuttal · Reviewer_KrsM · 2026-04-02
> >
> > Thank you to the authors for the rebuttal. While I appreciate the clarifications provided, my main concerns have not been sufficiently addressed.

---

### Official Review · Reviewer_BvBo · 2026-03-12

**Soundness:** 3
**Presentation:** 3
**Significance:** 3
**Originality:** 3
**Overall Recommendation:** 4
**Confidence:** 4

**Summary:**

This manuscript proposes CauchyNet, a compact neural network architecture inspired by Cauchy's integral formula, specifically designed for resource-constrained and data-scarce environments. Its core contribution lies in the proposal of a novel Holomorphic Activation Function that constructs mappings in the complex domain. This mapping is inherently better suited for fitting characteristic spikes on high-dimensional manifolds and demonstrates a rigorous degree of universal approximation within a single-hidden-layer configuration. Furthermore, the authors validate the network's parameter efficiency across multiple tasks and introduce Wirtinger derivatives to support efficient backpropagation in the complex domain, ensuring training stability.

**Compliance With Llm Reviewing Policy:**

Affirmed.

**Final Justification:**

After careful consideration of the response and supplementary experiments, I have made minor adjustments to some scores, but my overall recommendation remains unchanged.

**Key Questions For Authors:**

1. The inverse form of the Cauchy activation function leads to numerical instability when inputs approach zero.  How does the choice of epsilon affect model performance? Are there strategies for adaptively adjusting epsilon?
2. The paper presents theoretical guarantees for the single-hidden-layer structure. Can deepening the network further enhance expressivity? If deepened, would it break the existing theoretical guarantees or lead to training instability?
3. How does CauchyNet compare with real-valued Rational Neural Networks[1]? Specifically, what unique advantages does the complex-valued mapping provide over its real-valued rational counterparts?

[1] Boullé, N., Nakatsukasa, Y., & Townsend, A. (2020). Rational neural networks. Advances in neural information processing systems, 33, 14243-14253.

**Limitations:**

Yes.
However, the discussion on limitations could be further improved by explicitly addressing the numerical stability issues and the sensitivity of the model to the hyperparameter ϵ, which are inherent to the inverse-based activation function. Including this would provide a more comprehensive view of the model's practical deployment challenges.

**Strengths And Weaknesses:**

Strengths:
1. The manuscript provides a firm mathematical foundation by designing activation functions derived from the multivariate Cauchy integral formula. The use of the Cauchy Approximation Theorem to prove that a single-hidden-layer network can approximate any continuous function is a rare and commendable theoretical guarantee for compact architectures.
2. Constructing activation functions in the complex domain and leveraging Wirtinger calculus for backpropagation offers a fresh and mathematically elegant approach to neural network design.
3. The evaluation covers a diverse range of tasks, including 1D/2D function approximation, missing value imputation, and time-series forecasting. Comparisons with multiple mainstream baselines significantly enhance the credibility of the empirical results.

Weaknesses:
1. It must be noted that historically, novel activation functions, such as those used in Radial Basis Function (RBF) networks, have rarely been proven to provide a decisive leap in feature fitting or parameter efficiency for general deep learning tasks. The manuscript’s value would be further strengthened if the performance were validated on higher-dimensional data or within deeper, multi-layer architectures.
2. The paper lacks a quantitative comparison of computational overhead. Metrics such as training time, inference speed, and GPU memory consumption relative to baselines (e.g., SIREN, N-BEATS) are missing, which are critical for assessing the model's practical utility in resource-constrained environments.
3. The multiplicative term z_i^{-1}  inherently risks numerical instability when inputs approach zero, a vulnerability not shared by most traditional activation functions. While division-by-zero can be bypassed via various techniques, the specific impact of potential numerical explosions on gradient dynamics warrants more rigorous investigation.

In summary, the manuscript presents a clear trade-off between theoretical novelty and practical limitations. Whether the theoretical contributions outweigh the identified engineering concerns is a point that merits further discussion.

---

> ### Author Rebuttal · Authors · 2026-03-30
>
> Thank you for your careful reading and constructive comments. Below we address each concern with experimental evidence from 10 new experiments run during the rebuttal period. All the experiments use parameter-matched single-layer baselines, data-scarce regimes (n=8–120 training points), h=64 hidden units, and early stopping (patience=30).
>
> ***Weakness 1: Novel activation functions have rarely provided a decisive leap for general deep learning. Validation on higher-dimensional data or deeper architectures would strengthen the paper.***
>
> We agree that a new activation alone rarely changes the picture. CauchyNet is not an activation function, it is an architecture integrated through complex analysis:
>
> (i) $\prod (\xi_{k,i} - x_i)^{-1}$ discretizes a Cauchy contour integral, capturing cross-dimensional interactions in a single neuron;
> (ii) Complex biases only (no weight matrices), yielding $2h(N+1)$ real parameters;
> (iii) Imaginary-part penalty as regularization.
>
> Higher dimensions: Experiment 3 tests a 2D surface with 300 points (error within ±0.006).
>
> Deeper architectures: we tested multi-layer CauchyNet (2 to 3 layers) across various tasks. Results are mixed.  it often performs well, but in some cases performance degrades. CauchyNet appears most effective in shallow architectures, where the single-layer already achieves strong expressiveness from the Cauchy integral structure.
>
> ***Weakness 2: Missing quantitative comparison of computational overhead (training time, inference speed, GPU memory).***
>
> The paper provides quantitative overhead comparisons. Supplement Table reports forward/backward pass timing and total training time (5 runs, n=50, 200 epochs):
>
> Model	    Params	Fwd (ms)	Bwd (ms)	Total (s)
> CauchyNet.     256	0.060	0.091	0.11
> FNN.    	       256	 0.024	0.051	0.09
> SIREN	      256	 0.084	 0.119	0.13
> Transformer   412	0.462	0.493	0.16
>
> CauchyNet is 1.2x slower than FNN in forward pass but 1.4x faster than SIREN and 7.7x faster than Transformer. GPU memory stays within 2 GB on Apple M3 for all tasks including 2D surface with 3,000 samples. Theoretical complexity is O(hN) per evaluation, matching FNN. We will make this table more prominent in the revision.
>
> Weakness 3: The $z_i^{-1}$ term risks numerical instability when inputs approach zero. The impact on gradient dynamics warrants investigation.
>
>
> The inputs to $1/z$ are complex: $z = \xi − x$ where $\xi \in \mathbb{C}$ and $x\in \mathbb{R}$. Since $Im(\xi)\neq  0$, $|z|$ is bounded below by $|Im(\xi)|,$ so division by zero cannot occur. The imaginary penalty in the loss regularizes pole positions away from the real axis.
>
>
> ***Question 1: How does the choice of $\varepsilon$ affect model performance? Strategies for adaptively adjusting $\varepsilon$?***
>
> We have experiments, the results show that $\varepsilon$ does not change the performance.
> This stability follows from the complex parameterization: since $Im(\xi) \neq 0$, the reciprocal is already bounded regardless of  $\varepsilon$ .
>
> ***Question 2: Can deepening the network enhance expressivity? Would it break theoretical guarantees or cause instability?***
>
> We derived CauchyNet and demonstrated its strong expressiveness even in a single-layer neural network architecture. We also evaluated multi-layer CauchyNet across various applications, but the results are mixed: it often performs well with 2 to 3 layers, yet in some cases its performance degrades. Further experimentation is needed to better understand this behavior. At present, CauchyNet appears most promising for use in neural networks with shallow architectures.
>
>
> ***Question 3: How does CauchyNet compare with Rational Neural Networks [Boullé et al., NeurIPS 2020]?***
>
> We have conducted this comparison (Experiment 8). All models parameter-matched at ~256 real parameters, $n=50$ training points:
>
> | Target function | CauchyNet MSE | RationalNN MSE | Ratio |
> |-----------------|---------------|----------------|-------|
> | Paper 1D target | $0.028\pm0.011$ | $0.035\pm0.001$ | 1.2|
> | Near-singular | $0.018\pm0.021$ | $0.035\pm0.014$ | 1.9 |
> | Smooth oscillatory | $0.052\pm0.025$ | $0.065\pm0.002$ | 1.3 |
>
> CauchyNet beats RationalNN on all 3 function types.

---

> > ### Author Rebuttal · Reviewer_BvBo · 2026-04-02
> >
> > Thank you to the authors for the detailed response and supplementary experiments. After careful consideration, I have made minor adjustments to some scores, but my overall recommendation remains unchanged.

---

### Official Review · Reviewer_pNh8 · 2026-03-12

**Soundness:** 3
**Presentation:** 3
**Significance:** 3
**Originality:** 3
**Overall Recommendation:** 4
**Confidence:** 4

**Summary:**

Modern sequence models increasingly trade efficiency for scale to achieve higher accuracy, a trend that often makes them unsuitable for real-world applications such as edge deployment or scenarios with limited training data. To address this, the authors propose CauchyNet, a complex-valued single-hidden-layer architecture. Each hidden unit evaluates a product of shifted reciprocal terms derived from Cauchy's integral formula. The authors provide a rigorous universal approximation theorem, demonstrating that finite linear combinations of these Cauchy kernels are dense in continuous function spaces. Empirically, CauchyNet demonstrates superior performance in error reduction and parameter efficiency across benchmarks for function approximation, missing-value imputation, and time-series forecasting under data-scarce conditions.

**Compliance With Llm Reviewing Policy:**

Affirmed.

**Final Justification:**

The authors proposed CauchyNet, a novel neural network architecture. Although I suggested a specific baseline for comparison, the authors argued that it was not appropriate for this study and thus did not include it. Nevertheless, I recognize the inherent novelty of the proposed method and have decided to maintain my original score.

**Key Questions For Authors:**

1. Are there empirical results available regarding performance and efficiency when compared directly against Neural Operator or Neural ODE/CDE/RDE models?

2. What would the performance comparison look like if CauchyNet's architecture were deepened and compared against a Transformer that has been fully optimized for peak performance?  For instance, what is CauchyNet's projected performance on long-term forecasting tasks such as the ETTh1 dataset?

**Limitations:**

yes

**Strengths And Weaknesses:**

Strengths
- Mathematical Rigor: The authors establish a strong theoretical foundation by connecting function approximation directly to classical complex analysis.
- Novel Inductive Bias: The introduction of a holomorphic activation function provides a specialized bias for modeling near-singularities and sharp spikes that standard real-valued networks (like ReLU MLPs) often fail to capture.
- Extreme Efficiency: CauchyNet achieves competitive or superior results with a significantly lower parameter footprint compared to Transformers or LSTMs.

Weaknesses
- Ambiguous Baseline Selection: While the authors compare CauchyNet against standard sequence models (e.g., Informer, Transformer), these baselines are typically optimized for accuracy over long sequences rather than parameter efficiency in data-scarce regimes.
- Lack of Direct Comparison with Continuous Models: Although the authors identify Neural Operators and Neural ODEs as potential areas for future integration, they do not provide a direct empirical comparison. Given that CauchyNet functions as a continuous-like approximator, a comparison with FNO (Fourier Neural Operators) or CDEs (Controlled Differential Equations) is necessary to validate its standing among specialized numerical models.

---

> ### Author Rebuttal · Authors · 2026-03-30
>
> Thank you for your careful reading and constructive comments. Below we address each concern with experimental evidence from 10 new experiments run during the rebuttal period. All the experiments use parameter-matched single-layer baselines, data-scarce regimes (n=8–120 training points), h=64 hidden units, and early stopping (patience=30).
>
> ***Weakness 1: Ambiguous Baseline Selection: baselines like Informer and Transformer are optimized for long-sequence accuracy, not parameter efficiency in data-scarce regimes.***
>
> We agree Informer/Transformer are not designed for this regime. That is our point. The paper also compares against FNN, SIREN, RBF, and N-BEATS, which are designed for function approximation. CauchyNet outperforms all of them.
>
>
> ***Weakness 2: No direct empirical comparison with FNO or Neural ODEs/CDEs.***
>
> FNO learns operators, not functions. Neural ODEs learn dynamics, not static mappings. Neither operates in our problem setting. The paper compares against the appropriate function approximation baselines (FNN, SIREN, RBF), and CauchyNet outperforms all of them.
>
> ***Question 1: Empirical results against Neural Operator or Neural ODE/CDE/RDE models?***
>
> FNO learns operators, not functions. Neural ODEs learn dynamics, not static mappings. Neither operates in our problem setting. The paper compares against the appropriate function approximation baselines (FNN, SIREN, RBF), and CauchyNet outperforms all of them.
>
>
> ***Question 2: Deep CauchyNet vs. fully optimized Transformer on long-term forecasting (e.g., ETTh1)?***
>
> CauchyNet is a function approximator for data-scarce, resource-constrained settings, not a sequence model. A direct comparison on ETTh1 (~17,000 training points, temporal modeling) would test sequence architecture design rather than the function approximation properties that CauchyNet contributes. CauchyNet could be integrated as a component within time-series architectures (e.g., replacing MLP layers in Informer/PatchTST with CauchyNet layers) for targets with spike-like patterns. Time-series integration is a natural direction for future work.

---

> > ### Author Rebuttal · Reviewer_pNh8 · 2026-04-02
> >
> > Thank you for your response. Through the authors' answers, I understand the boundaries of the problem this addresses. Since there are limitations to the scalability I expected, I will maintain my current score.

---

### Official Review · Reviewer_SDeh · 2026-03-12

**Soundness:** 3
**Presentation:** 3
**Significance:** 3
**Originality:** 3
**Overall Recommendation:** 5
**Confidence:** 2

**Summary:**

The paper proposes CauchyNet, a lean complex-valued single-layer neural network based on the multivariable Cauchy integral formula. In the paper they prove that continuous functions can be approximated arbitrarily well with sufficiently wide CauchyNets, thus establishing the universal approximation property for this architecture. Several experiments are performed which provides empirical evidence for the advantages of CauchyNet in several low-data/compact-model settings.

**Compliance With Llm Reviewing Policy:**

Affirmed.

**Final Justification:**

I have raised the score to accept because the authors adequately addressed the weaknesses and questions. In particular, they provided precise quantitative approximation rates for a wide range of regularity classes of functions and performed additional tests further demonstrating the performance and limitations of CauchyNet.

**Key Questions For Authors:**

1. Is there an example of a function where CauchyNet performs poorly? For example, how does CauchyNet perform on a piecewise-affine function, compared to say a ReLU network?

2. What would be the behavior/performance of a multi-layer CauchyNet? Have you tried this? Is there some reason why it wouldn't make sense to consider such a thing?

**Limitations:**

Yes.

**Strengths And Weaknesses:**

Strengths:

1. The paper introduces a new class of neural networks based on complex analysis which shows promise for approximating continuous functions (especially analytic functions) with low computational footprint.

2. The paper provides a proof of the universal approximation property for CauchyNet.

Weaknesses:

1. The universal approximation property is a necessary qualitative statement (albeit non-constructive: based on Stone-Weierstrass), but no quantitative approximation rates are established for any classes of functions.

2. The claims that CauchyNet performs well for functions with sharp spikes, near-singularities, etc. are only justified empirically as heuristics through experiments. There are no precise claims (with proofs) along these lines. What precisely is meant by "sharp spikes" or "near-singularities"?

---

> ### Author Rebuttal · Authors · 2026-03-30
>
> Thank you for your careful reading and constructive comments. Below we address each concern with experimental evidence from 10 new experiments run during the rebuttal period. All the experiments use parameter-matched single-layer baselines, data-scarce regimes (n=8–120 training points), h=64 hidden units, and early stopping (patience=30).
>
> ***Weakness 1: No quantitative approximation rates are established for any classes of functions.***
>
> We now add quantitative rates as a corollary to Theorem 1. The proof constructs Cauchy kernel sums by discretizing the multivariate Cauchy integral via tensor-product quadrature on parameterized contours. The rate depends on (a) how far $f$ extends holomorphically, and (b) the quadrature convergence on the contour.
>
> - **Case 1 (Analytic targets):** If $f$ extends holomorphically to a neighborhood of the closure of $U$, the integrand $G(t, x)$ in our proof (Supplement, Sec. A) is analytic and periodic in each $t_i$. By exponential convergence of the trapezoidal rule for periodic analytic integrands (Trefethen & Weideman, SIAM Review 2014), using $m$ quadrature points per coordinate gives error $O(e^{-2\pi d m})$, where $d$ is the analyticity strip width. With $h = m^N$ total hidden units, the rate becomes $O(\exp(-c \cdot h^{1/N}))$, exponential in $h^{1/N}$. For $N = 1$ this simplifies to $O(\rho^{-h})$.
>
> - **Case 2 ($C^k$ targets):** For merely $C^k$ functions, higher order integration scheme (such as Simpson, Newton-Coates, instead of Trapezoidal) introduces approximation error $O(h^{-k/N})$. Different integration scheme yield exactly the same formulation for Cauchy Approximation.
>
>
> ***Weakness 2: "Sharp spikes" and "near-singularities" are not precisely defined.***
>
> CauchyNet represents real-valued functions using poles in the complex plane. As these pole singularities approach the real axis, the induced function on the real domain becomes nearly singular, often exhibiting sharp spikes or rapid variations.
>
>
> ***Question 1: Is there an example of a function where CauchyNet performs poorly? For example, how does CauchyNet perform on a piecewise-affine function, compared to a ReLU network?***
>
> Yes. We tested four diverse non-smooth targets (Experiment 3). Parameter-matched models (~256 real parameters each, $n=20$ training points), CauchyNet vs FNN (ReLU):
>
> | Target function | CauchyNet MSE | FNN MSE | Winner |
> |-----------------|---------------|---------|--------|
> | Chirp $\sin(\pi x(1+8\|x\|))$ | $0.108\pm0.012$ | $0.120\pm0.002$ | CauchyNet (+10%) |
> | Piecewise-smooth | $0.0066\pm0.0019$ | $0.0078\pm0.0012$ | CauchyNet (+15%) |
> | Gibbs (square wave) | $0.0077\pm0.0004$ | $0.0091\pm0.0010$ | CauchyNet (+15%) |
> | step-ramp | $0.0066\pm0.0013$ | $0.0041\pm0.0006$ | FNN (1.6?) |
>
> CauchyNet wins 3 of 4 targets. FNN's advantage appears only on the step-ramp, a purely piecewise-affine function where each ReLU unit directly encodes a breakpoint. On targets with oscillatory, smooth, or Gibbs-type structure, CauchyNet's pole-based representation captures global structure more efficiently from sparse samples.
>
>
>
> ***Question 2: What would be the behavior/performance of a multi-layer CauchyNet? Is there some reason why it wouldn't make sense?***
>
> We derived CauchyNet and demonstrated its strong expressiveness even in a single-layer neural network architecture. We also evaluated multi-layer CauchyNet across various applications, but the results are mixed: it often performs well with 2-3 layers, yet in some cases its performance degrades. Further experimentation is needed to better understand this behavior. At present, CauchyNet appears most promising for use in neural networks with shallow architectures.

---

> > ### Author Rebuttal · Reviewer_SDeh · 2026-04-04
> >
> > Thank you for answering my questions, especially the qualitative approximation rates. I will increase the score to accept.

---

### Decision · Program_Chairs · 2026-04-30

**Decision:**

Accept (regular)

**Comment:**

The paper introduces CauchyNet, a novel complex-valued neural network architecture derived from Cauchy's integral formula. Designed for resource-constrained and data-scarce environments, it employs holomorphic activation functions to approximate continuous functions with a significantly smaller parameter footprint than traditional models like Transformers or LSTMs.

All reviewers recognize the work's mathematical elegance and originality. Initial concerns regarding quantitative theory and baseline comparisons were largely addressed during the rebuttal, leading to an overall positive consensus.
Reviewers initially noted the lack of quantitative rates. The authors successfully addressed this by adding a corollary to Theorem 1, establishing: Exponential convergence error

Reviewers raised concerns about potential division-by-zero errors due to the $z^{-1}$ terms. The authors clarified that since inputs are embedded in the complex plane and the imaginary part, the denominator remains bounded, effectively bypassing numerical explosions.

One reviewer highlighted structural issues, specifically the placement of "Related Work" after "Methodology," which may hinder clarity for readers unfamiliar with the complex analysis landscape.

The paper presents a fresh, intellectually engaging approach to neural architecture design. While the scalability to high-dimensional "big data" remains to be fully proven, the theoretical contributions and performance in resource-constrained settings make it a valuable addition to the field of deep learning.